# Multiple factors maintain assembled trans-SNARE complexes in the presence of NSF and αSNAP

Eric A Prinslow[1,2,3], Karolina P Stepien[1,2,3], Yun-Zu Pan[1,2,3], Junjie Xu[1,2,3], Josep Rizo[1,2,3]*

[1]Department of Biophysics, University of Texas Southwestern Medical Center, Dallas, United States; [2]Department of Biochemistry, University of Texas Southwestern Medical Center, Dallas, United States; [3]Department of Pharmacology, University of Texas Southwestern Medical Center, Dallas, United States

**Abstract** Neurotransmitter release requires formation of trans-SNARE complexes between the synaptic vesicle and plasma membranes, which likely underlies synaptic vesicle priming to a release-ready state. It is unknown whether Munc18-1, Munc13-1, complexin-1 and synaptotagmin-1 are important for priming because they mediate trans-SNARE complex assembly and/or because they prevent trans-SNARE complex disassembly by NSF-αSNAP, which can lead to de-priming. Here we show that trans-SNARE complex formation in the presence of NSF-αSNAP requires both Munc18-1 and Munc13-1, as proposed previously, and is facilitated by synaptotagmin-1. Our data also show that Munc18-1, Munc13-1, complexin-1 and likely synaptotagmin-1 contribute to maintaining assembled trans-SNARE complexes in the presence of NSF-αSNAP. We propose a model whereby Munc18-1 and Munc13-1 are critical not only for mediating vesicle priming but also for precluding de-priming by preventing trans-SNARE complex disassembly; in this model, complexin-1 also impairs de-priming, while synaptotagmin-1 may assist in priming and hinder de-priming.

DOI: https://doi.org/10.7554/eLife.38880.001

*For correspondence:
Jose.Rizo-Rey@UTSouthwestern.edu

**Competing interests:** The authors declare that no competing interests exist.

## Introduction

The release of neurotransmitters by $Ca^{2+}$-triggered synaptic vesicle exocytosis is a central event for interneuronal communication and involves multiple steps. Synaptic vesicles first dock at specialized sites on the plasma membrane called active zones, undergo one or more priming reactions that leave the vesicles ready for release, and fuse with the plasma membrane upon $Ca^{2+}$ influx evoked by an action potential (*Südhof, 2013*). Extensive research has shown that these steps are exquisitely regulated by a sophisticated protein machinery and has led to defined models for the functions of key components from this machinery (*Brunger et al., 2018*; *Jahn and Fasshauer, 2012*; *Rizo, 2018*; *Südhof and Rothman, 2009*). The soluble N-ethylmaleimide sensitive factor attachment protein receptors (SNAREs) synaptobrevin, syntaxin-1 and SNAP-25 form a parallel four-helix bundle called the SNARE complex that brings the synaptic vesicle and plasma membranes together and is key for membrane fusion (*Hanson et al., 1997*; *Poirier et al., 1998*; *Söllner et al., 1993*; *Sutton et al., 1998*). N-ethylmaleimide sensitive factor (NSF) and soluble NSF attachment proteins (SNAPs; no relation to SNAP-25) disassemble SNARE complexes after release to recycle the SNAREs for another round of fusion (*Banerjee et al., 1996*; *Mayer et al., 1996*; *Söllner et al., 1993*). Munc18-1 and Munc13s orchestrate SNARE complex assembly in an NSF-SNAP-resistant manner (*Ma et al., 2013*) that improves the fidelity of parallel assembly (*Lai et al., 2017*). The underlying mechanism involves binding of Munc18-1 to a self-inhibited 'closed conformation' of syntaxin-1 (*Dulubova et al., 1999*; *Misura et al., 2000*) and to synaptobrevin, thus forming a template to assemble the SNARE complex

(*Baker et al., 2015*; *Parisotto et al., 2014*; *Sitarska et al., 2017*), while Munc13s bridge the vesicle and plasma membranes (*Liu et al., 2016*) and help to open syntaxin-1 (*Ma et al., 2011*; *Richmond et al., 2001*; *Yang et al., 2015*). Synaptotagmin-1 acts as the major $Ca^{2+}$ sensor that triggers release through interactions with phospholipids (*Fernández-Chacón et al., 2001*) and the SNARE complex (*Brewer et al., 2015*; *Zhou et al., 2015*; *Zhou et al., 2017*), in a tight interplay with complexins (*Giraudo et al., 2006*; *Schaub et al., 2006*; *Tang et al., 2006*).

Despite these and other crucial advances, fundamental questions remain about how trans-SNARE complexes that bridge the vesicle and plasma membranes are formed, about the interplay between Munc18-1, Munc13-1, NSF and αSNAP in promoting trans-SNARE complex assembly or disassembly, and about the nature of the primed state(s) of the release machinery in the readily-releasable pool (RRP) of vesicles. The primed state is believed to include trans-SNARE complexes that are partially formed, with the N-terminal half assembled and at least part of the C-terminal, membrane proximal portion unassembled (e.g. *Sørensen et al., 2006*; *Walter et al., 2010*), but it is unclear which other components are bound to the SNAREs in this state. Reconstitution assays showing that the fusion between synaptobrevin-containing liposomes and syntaxin-1-SNAP-25-containing liposomes observed in the presence of synaptotagmin-1 and $Ca^{2+}$ is abolished by NSF-αSNAP, but occurs efficiently upon further addition of Munc18-1 and a Munc13-1 fragment, led to the notion that Munc18-1 and Munc13-1 mediate a pathway for trans-SNARE complex assembly that is resistant to NSF-αSNAP (*Liu et al., 2016*; *Ma et al., 2013*), explaining the essential nature of Munc18-1 and Munc13s for vesicle priming (*Aravamudan et al., 1999*; *Richmond et al., 1999*; *Varoqueaux et al., 2002*; *Verhage et al., 2000*). This interpretation arose in part because NSF-αSNAP disassemble not only cis-SNARE complexes but also syntaxin-1-SNAP-25 heterodimers (*Hayashi et al., 1995*), thus preventing trans-SNARE complex formation by the SNAREs alone, and because of evidence suggesting that NSF-αSNAP cannot disassemble trans-SNARE complexes (*Weber et al., 2000*). However, studies of yeast vacuolar fusion showed that the NSF-αSNAP homologues Sec18-Sec17 disassemble trans-SNARE complexes and that disassembly is prevented by HOPS, a tethering complex that includes the Munc18-1 homologue Vps33 and coordinates SNARE complex formation (*Mima et al., 2008*; *Xu et al., 2010*). Moreover, recent reports showed that at least a fraction of neuronal trans-SNARE complexes can be disassembled by NSF-αSNAP in vitro (*Yavuz et al., 2018*) and that Munc18-1 and Munc13-1 are critical to prevent de-priming of readily-releasable synaptic vesicles in neurons, but such requirement can be bypassed by the NSF inactivating agent N-ethylmaleimide (*He et al., 2017*).

These findings suggest that the cytoplasmic environment of a presynaptic terminal favors disassembly of all kinds of SNARE complexes and hence that the trans-SNARE complexes formed after priming must be protected against disassembly by NSF-αSNAP. However, the mechanisms underlying such protection are unknown. The results of *He et al. (2017)* indicate that Munc18-1 and Munc13-1 play key roles in such protection, in addition to mediating an NSF-αSNAP-resistant pathway of trans-SNARE complex assembly, but this hypothesis has not been tested, and Munc18-1 and Munc13-1 are often assumed to be dispensable after mediating assembly. Moreover, it is plausible that protection against disassembly by NSF-αSNAP depends also on other proteins such as synaptotagmin-1 and complexins, which bind to SNARE complexes and have also been proposed to facilitate trans-SNARE complex formation (*Diao et al., 2013*; *Li et al., 2017*). In this context, while initial studies suggested that synaptic vesicle priming is not altered in neurons from synaptotagmin-1 knockout (KO) mice (*Geppert et al., 1994*) and in complexin-1/2 double knockout (DKO) mice (*Reim et al., 2001*), subsequent analyses revealed that absence of these proteins does decrease the RRP of vesicles (*Bacaj et al., 2015*; *Chang et al., 2018*; *Xue et al., 2010*; *Yang et al., 2010*). Such decreases were not as dramatic as those observed in Munc18-1 KO mice (*Verhage et al., 2000*) and Munc13-1/2 DKO mice (*Varoqueaux et al., 2002*), where priming is totally abrogated, but do suggest that synaptotagmin-1 and complexins are involved in priming or perhaps in maintenance of the RRP. Thus, while it is known that the existence of an RRP of vesicles depends on Munc18-1, Munc13-1, synaptotagmin-1 and complexins, it is still unclear to what extent the roles of these various factors arise because they mediate priming by facilitating trans-SNARE complex assembly and/or because they stabilize primed vesicles by protecting against trans-SNARE complex disassembly by NSF-SNAPs.

The study presented herein was designed to address these questions and understand the interplay between these proteins in trans-SNARE complex assembly and disassembly, using a

fluorescence resonance energy transfer (FRET) assay. Our data show that trans-SNARE complex assembly in the presence of NSF-αSNAP requires Munc18-1 and Munc13-1, as expected from our previous reconstitution experiments (*Ma et al., 2013*), but does not require complexin-1. Moreover, we find that Munc18-1 and Munc13-1 synergistically help to maintain assembled trans-SNARE complexes in the presence of NSF-αSNAP, which is strongly enhanced by Ca$^{2+}$, and that trans-SNARE complexes are protected against disassembly by complexin-1. Synaptotagmin-1 facilitates NSF-αSNAP-resistant trans-SNARE complex assembly and may contribute to stabilizing trans-SNARE complexes, but its effects are less marked. We propose a model whereby Munc18-1 and Munc13-1 play key roles not only in priming synaptic vesicles to a readily-releasable state but also in protecting them against de-priming by NSF-SNAPs, while synaptotagmin-1 plays a less critical role in both priming and maintenance of the RRP, and complexin-1 does not mediate priming but stabilizes primed vesicles.

## Results

### A sensitive assay to monitor trans-SNARE complex assembly and disassembly

In order to investigate the factors that influence trans-SNARE complex assembly and disassembly, membrane fusion must be prevented to avoid the conversion of trans-SNARE complexes into cis complexes that are well known to be disassembled by NSF-αSNAP (*Söllner et al., 1993*). To set up a trans-SNARE complex assembly assay without interference from membrane fusion, we used a similar approach to that described recently by *Yavuz et al. (2018)*, which was published during the course of this work and used a mutation at the C-terminus of the synaptobrevin SNARE motif to prevent C-terminal assembly of the SNARE complex. For our assay, we designed a SNAP-25 mutant bearing two single residue substitutions (M71D,L78D) that replace buried hydrophobic residues with negatively charged residues at the C-terminus of the SNARE four-helix bundle (*Figure 1A*), and thus are also expected to strongly hinder C-terminal zippering of the SNARE complex. To verify this expectation, we used a membrane fusion assay that simultaneously measures lipid and content mixing between synaptobrevin-containing liposomes and syntaxin-1-SNAP-25-containing liposomes in the presence of NSF, αSNAP, Munc18-1 and a fragment containing the C$_1$, C$_2$B, MUN and C$_2$C domains of Munc13-1 (*Liu et al., 2016*; *Liu et al., 2017*). This fragment, which we refer to as C$_1$C$_2$BMUNC$_2$C, spans the entire highly conserved C-terminal region of Munc13-1 and is sufficient to efficiently rescue neurotransmitter release in Munc13-1/2 DKO neurons (*Liu et al., 2016*). As expected, we observed highly efficient, Ca$^{2+}$-dependent membrane fusion in experiments performed with wild type (WT) SNAP-25; however, content mixing was abolished and lipid mixing was very inefficient when SNAP-25m was used in the assays instead of WT SNAP-25 (*Figure 1B,C*), demonstrating that the M71D,L78D mutation in SNAP-25 indeed prevents membrane fusion. Note that the small amount of lipid mixing that we observed might arise from lipid transfer without membrane merger (*Rizo, 2018*) and that some lipid mixing was observed in reconstitution assays even when long flexible linkers were introduced between the synaptobrevin SNARE motif and transmembrane region (*McNew et al., 1999*).

To test for formation of trans-SNARE complexes, we developed a FRET assay based on attachment of donor (Alexa488) and acceptor (tetramethylrhodamine, TMR) fluorescent probes on single-cysteine mutants of full-length synaptobrevin (L26C) and syntaxin-1 (S186C), respectively (all native cysteines were mutated to serine or hydrophobic residues). Residues L26 of synaptobrevin and S186 of syntaxin-1 were chosen to place the fluorescent probes because they closely precede the N-termini of the SNARE motifs in the four-helix bundle (*Sutton et al., 1998*), that is residue 29 of synaptobrevin and 190 of syntaxin-1 (*Figure 1A*). Hence, SNARE complex assembly is not expected to be perturbed by attachment of fluorescent probes to these residues but should bring the fluorescent probes into close proximity for efficient FRET, while disassembly by NSF-αSNAP should eliminate the FRET (*Figure 2A*). Attachment of fluorescent probes to residue 26 of synaptobrevin and 186 of syntaxin-1 is also expected to have no effect on binding of both proteins to NSF-αSNAP, complexin-1, synaptotagmin-1 or Munc18-1 based on the three-dimensional structural information available on the 20S complex formed by NSF, αSNAP and the SNAREs (*Zhao et al., 2015*), on the complexin-SNARE complex (*Chen et al., 2002*), on three synaptotagmin-1-SNARE complexes (*Brewer et al.,*

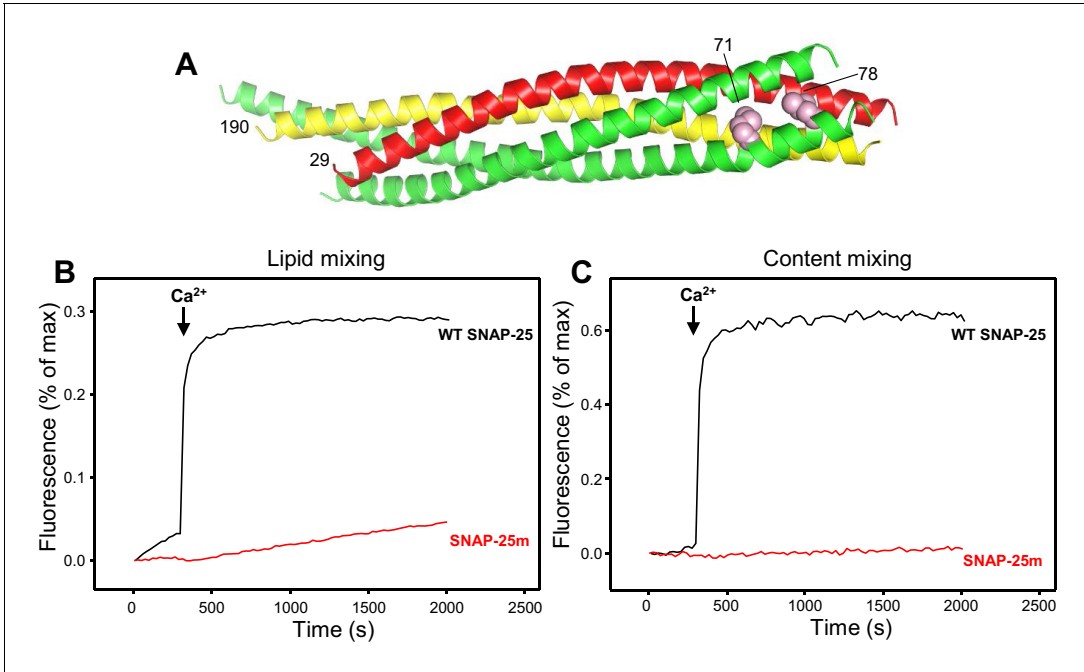

**Figure 1.** Design of a SNAP-25 mutation that abrogates its ability to support membrane fusion. (**A**) Ribbon diagram of the crystal structure of the SNARE complex (PDB accession code 1SFC) (*Sutton et al., 1998*). Synaptobrevin is red, syntaxin-1 yellow and SNAP-25 green, with the side chains of the two residues that were mutated to aspartate (M71 and L78) shown as pink spheres. Note that the side chains are pointing toward the hydrophobic interior of the four-helix bundle. Hence, mutating these residues to aspartate is expected to prevent C-terminal zippering of the SNARE complex. The residue numbers of the two mutated residues and of the N-termini of synaptobrevin and syntaxin-1 SNARE motifs are indicated. (**B,C**) The SNAP-25 M71D,L78D mutation abrogates membrane fusion in reconstitution assays. Lipid mixing (**B**) between V- and T-liposomes was monitored from the fluorescence de-quenching of Marina Blue lipids and content mixing (**C**) was monitored from the increase in the fluorescence signal of Cy5-streptavidin trapped in the V-liposomes caused by FRET with PhycoE-biotin trapped in the T-liposomes upon liposome fusion. The assays were performed in the presence of Munc18-1, Munc13-1 $C_1C_2BMUNC_2C$, NSF and $\alpha$SNAP with T-liposomes that contained syntaxin-1 and wild type (WT) SNAP-25 or SNAP-25 M71D,L78D mutant (SNAP-25m). Experiments were started in the presence of 100 µM EGTA and 5 µM streptavidin, and $Ca^{2+}$ (600 µM) was added at 300 s.

DOI: https://doi.org/10.7554/eLife.38880.002

*2015*; *Zhou et al., 2015*; *Zhou et al., 2017*), on the Munc18-1-closed syntaxin-1 complex (*Misura et al., 2000*), and on the vacuolar Vps33-Nyv1 complex (*Baker et al., 2015*), which most likely provides a reliable model for the homologous Munc18-1-synaptobrevin complex (*Sitarska et al., 2017*). Since in the experiments described below we relied on the donor fluorescence emission to monitor FRET, we tested whether the emission spectrum of liposomes containing Alexa488-synaptobrevin is affected by various proteins used in this study, including Munc18-1, Munc13-1 $C_1C_2BMUNC_2C$, NSF, $\alpha$SNAP, complexin-1 and a soluble fragment of synaptotagmin-1 spanning the two $C_2$ domains that form most of its cytoplasmic region ($C_2AB$) and include its $Ca^{2+}$-binding sites (*Fernandez et al., 2001*; *Sutton et al., 1995*; *Ubach et al., 1998*). None of these proteins substantially affected the fluorescence spectrum except for a slight increase in fluorescence caused by Munc13-1 $C_1C_2BMUNC_2C$ (*Figure 2—figure supplement 1*) that did not affect the conclusions derived from our data.

A potential problem with FRET assays to monitor trans-SNARE complex formation is that only a small subset of the SNAREs may form these complexes, leading to low FRET efficiency and hindering quantification of the degree of SNARE complex assembly (or disassembly). To maximize the amount of observable FRET based on the decrease in donor fluorescence emission intensity, we employed liposomes containing Alexa488-labeled synaptobrevin at a low protein-to-lipid (P/L) ratio (1:10,000) (V-liposomes) and used a large excess of liposomes containing TMR-syntaxin-1 and SNAP-25m at higher P/L ratio (1:800) (T-liposomes) in our FRET assays. Mixing the V- and T-liposomes at a 1:4 ratio led to a very slow decrease in donor fluorescence intensity (*Figure 2—figure supplement 2A*) that shows that trans-SNARE complex assembly is very inefficient under these conditions, most likely

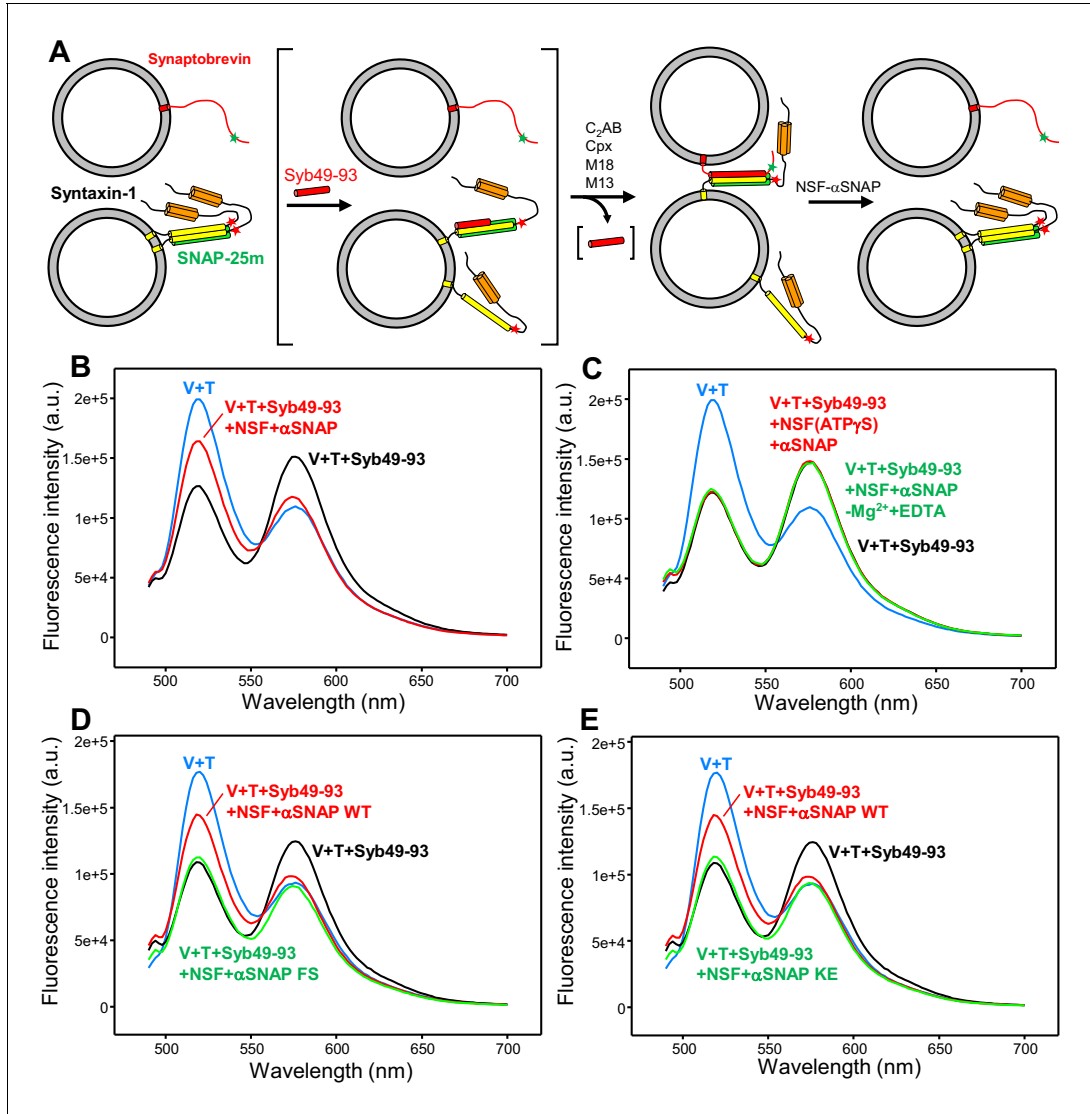

**Figure 2.** An assay to measure assembly of trans-SNARE complexes and disassembly by NSF-αSNAP. (**A**) Diagram illustrating the assay used to monitor trans-SNARE complex assembly and disassembly. V-liposomes containing synaptobrevin labeled with a FRET donor (Alexa488, green star) at residue 26 are mixed with T-liposomes containing SNAP-25m and syntaxin-1 labeled at residue 186 with a FRET acceptor (TMR, red star) in the presence of different factors. After monitoring the decrease in donor fluorescence intensity resulting from trans-SNARE complex formation under diverse conditions, NSF and αSNAP are added to test for disassembly of trans SNARE complexes. Synaptobrevin is red, SNAP-25m green and syntaxin-1 orange (N-terminal $H_{abc}$ domain) and yellow (SNARE motif). Although an excess of SNAP-25m was used in preparing the syntaxin-1-SNAP-25m liposomes, the majority of syntaxin-1-SNAP-25m complexes are expected to have a 2:1 stoichiometry such that the second syntaxin-1 SNARE molecule occupies the position of the synaptobevin SNARE motif in the SNARE four-helix bundle (bottom left diagram), hindering SNARE complex formation (reviewed in *Rizo and Südhof, 2012*). In some of the experiments, trans-SNARE complex assembly was facilitated by inclusion of the Syb49-93 peptide, which spans the C-terminal part of the synaptobrevin SNARE motif and displaces the second syntaxin-1 molecule from the syntaxin-1-SNAP-25m heterodimer, yielding the intermediate shown between brackets. Because Syb49-93 lacks the N-terminal half of the synaptobrevin SNARE motif, it can readily be displaced by full-length synaptobrevin to form trans-SNARE complexes (*Pobbati et al., 2006*). In other experiments, NSF-αSNAP were added from the beginning to investigate trans-SNARE complex assembly in their presence. (**B**) Fluorescence emission spectra (excitation at 468 nm) of a mixture of V-liposomes containing Alexa488-synaptobrevin and T-liposomes containing TMR-syntaxin-1-SNAP-25m (1:4 V- to T-liposome ratio) that had been incubated for five hours with Syb49-93 (black trace), and of the same sample after adding NSF-αSNAP plus ATP and $Mg^{2+}$ (red trace). The blue curve shows a control spectrum obtained by adding spectra acquired separately for V- and T-liposomes at the same concentrations. (**C**) Fluorescence emission spectra acquired under conditions similar to those of (**B**), with pre-incubated mixtures of Syb49-93 with V- and T-liposomes before (black curve) or after addition of NSF-αSNAP plus ATP and EDTA (green curve) or NSF-αSNAP plus ATPγS and $Mg^{2+}$ (red curve). (**D,E**) Fluorescence emission spectra acquired under similar conditions to those of (**B**), except that for the green curve WT αSNAP was replaced with the αSNAP FS (**D**) or KE (**E**) mutant. The red, black and blue curves are the same as in panel (**B**). All spectra were corrected for dilution caused by addition of reagents.

*Figure 2 continued on next page*

*Figure 2 continued*

DOI: https://doi.org/10.7554/eLife.38880.003

The following figure supplements are available for figure 2:

**Figure supplement 1.** Control experiments acquired to assess the effects of various factors on the fluorescence emission spectra of V-liposomes containing Alexa488-synaptobrevin.

DOI: https://doi.org/10.7554/eLife.38880.004

**Figure supplement 2.** Syb49-93 strongly accelerates trans-SNARE complex assembly.

DOI: https://doi.org/10.7554/eLife.38880.005

**Figure supplement 3.** Representative cryo-EM image of a mixture of V- and T-liposomes (1:4 ratio) that had been incubated with Syb49-93 for 5 hours before rapid freezing (scale bar, 100 nm).

DOI: https://doi.org/10.7554/eLife.38880.006

**Figure supplement 4.** Additional supporting fluorescence emission spectra.

DOI: https://doi.org/10.7554/eLife.38880.007

because SNARE complex assembly is hindered by formation of syntaxin-1-SNAP-25m heterodimers where synaptobrevin is replaced by a second syntaxin-1 molecule, leading to a 2:1 stoichiometry (*Figure 2A*). To overcome this problem, we used a synaptobrevin peptide spanning residues 49–93 (Syb49-93), which is expected to facilitate SNARE complex formation by displacing the second syntaxin-1 molecule (*Pobbati et al., 2006*). Indeed, inclusion of Syb49-93 strongly accelerated the rate of decrease in donor fluorescence intensity upon mixing V- and T-liposomes (*Figure 2—figure supplement 2A*). No further decrease in FRET was observed after five hours of incubation at 37°C, indicating that this time was sufficient to maximize the formation of trans-SNARE complexes. Cryo-electron microscopy (cryo-EM) images obtained for samples prepared under these conditions revealed well-dispersed liposomes that often exhibited close contacts with one or two other liposomes but did not form large clusters (*Figure 2—figure supplement 3*), as expected because of the use of low synaptobrevin-to-lipid ratios in the V-liposomes.

Comparison of the fluorescence emission spectrum acquired after incubating V- and T-liposomes with Syb49-93 for five hours with a control spectrum obtained by adding the spectra of separate samples of V- and T-liposomes confirmed a clear decrease in donor fluorescence, showing that efficient FRET developed as a result of trans-SNARE complex formation (*Figure 2B*, black and blue curves, respectively). The efficient FRET suggests that most of the accessible synaptobrevin molecules were incorporated into SNARE complexes, as only half of the Alexa488-labeled synaptobrevin molecules are expected to be accessible on the surface of the liposomes. Further addition of NSF-α SNAP led to a substantial but not complete recovery of the donor fluorescence (*Figure 2B*, red curve). These results show that NSF-αSNAP disassembled a fraction of the trans-SNARE complexes (estimated to be about 50%) while the remaining complexes were resistant to NSF-αSNAP, in agreement with the results of *Yavuz et al. (2018)*. It is worth noting that in these experiments there was a small degree of direct excitation of the large excess of acceptor probes used (see *Figure 2—figure supplement 4A*), even though the excitation wavelength corresponded to the donor. As expected, the acceptor fluorescence increased with respect to the V + T control upon incubating V- and T-liposomes for five hours, due to trans-SNARE complex formation (*Figure 2B*, black and blue curves). However, the acceptor fluorescence exhibited only a small increase upon addition of NSF-αSNAP (*Figure 2B*, red curve). This finding arises because the acceptor fluorescence is considerably affected by NSF-αSNAP, in contrast to the donor fluorescence (see below and *Figure 5—figure supplements 2* and *3*). Hence, the donor fluorescence provides a more reliable parameter than the acceptor fluorescence to assess the degree of trans-SNARE complex disassembly by NSF-αSNAP.

Disassembly of trans-SNARE complexes required ATP hydrolysis by NSF, as no disassembly was observed when $Mg^{2+}$ was replaced by EDTA in the reaction, or NSF was bound to ATPγS rather than ATP (*Figure 2C*). A similar amount of disassembly was observed in parallel assays performed with WT SNAP-25 instead of SNAP-25m in the presence of ATP and $Mg^{2+}$ (*Figure 2—figure supplement 4B*), showing that the reaction is not affected by the M71D,L78D mutation. We also examined whether trans-SNARE complex disassembly is affected by a K122E,K163E (KE) mutation in αSNAP that impairs SNARE binding (*Zhao et al., 2015*) and by an F27S,F28S (FS) mutation in an N-terminal loop of αSNAP that impairs disassembly of membrane-anchored cis-SNARE complexes because it disrupts binding of αSNAP to membranes (*Winter et al., 2009*). Both αSNAP mutations strongly

impaired the recovery of donor fluorescence observed when trans-SNARE complexes were disassembled by NSF in the presence of wild type (WT) αSNAP (compare green and red curves in *Figure 2D,E*), showing that interactions of αSNAP with both the membranes and the SNAREs are critical for trans-SNARE complex disassembly.

## Interplay between NSF, αSNAP, Munc18-1, Munc13-1, synaptotagmin-1 and complexin-1 in trans-SNARE complex assembly-disassembly

Our previous reconstitution studies showing that fusion between synaptobrevin-liposomes and syntaxin-1-SNAP-25-liposomes in the presence of NSF-αSNAP requires Munc18-1 and Munc13-1 led us to propose that Munc18-1 and Munc13-1 organize trans-SNARE complex assembly in an NSF-α SNAP resistant manner (*Liu et al., 2016*; *Ma et al., 2013*). However, trans-SNARE complex assembly was not directly monitored in these studies and it was unclear whether Munc18-1 and Munc13-1 were dispensable after trans-SNARE complex assembly. The finding that trans-SNARE complexes can be disassembled by NSF-αSNAP raises the question as to whether, in addition to providing an NSF-αSNAP-resistant pathway for trans-SNARE complex assembly, Munc18-1 and Munc13-1 protect assembled trans-SNARE complexes from disassembly by NSF-αSNAP. To address this question and also investigate the roles of synaptotagmin-1 and complexin-1 in trans-SNARE complex assembly and protection against disassembly, we performed kinetic experiments where we used our FRET assay, monitoring the decrease in donor emission fluorescence associated with trans-SNARE complex assembly in the presence of NSF-αSNAP and various combinations of Munc18-1, Munc13-1 $C_1C_2BMUNC_2C$, the synaptotagmin-1 $C_2AB$ fragment and complexin-1. $Ca^{2+}$ was added after 750 s to test its effects on assembly.

In the presence of Munc18-1 and Munc13-1 $C_1C_2BMUNC_2C$, we observed some slow trans-SNARE complex assembly before $Ca^{2+}$ addition and assembly was dramatically enhanced by $Ca^{2+}$, while there was almost no assembly in reactions with Munc18-1 alone or Munc13-1 $C_1C_2BMUNC_2C$ alone (*Figure 3A*). Complexin-1 and synaptotagmin-1 $C_2AB$ were unable to support trans-SNARE complex assembly in the presence of NSF-αSNAP even after $Ca^{2+}$ addition, and did not appear to enhance the rate of trans-SNARE complex assembly supported by Munc18-1 and Munc13-1 $C_1C_2BMUNC_2C$ (*Figure 3B*). These results confirm our proposal that Munc18-1 and Munc13-1 organize trans-SNARE complex assembly in an NSF-αSNAP resistant manner based on liposome fusion assays (*Liu et al., 2016*; *Ma et al., 2013*) but note that, in those assays, fusion might ensue quickly, in a concerted fashion, upon trans-SNARE complex formation without a chance for disassembly. Because in our FRET assays of trans-SNARE complex formation fusion is prevented by the mutation in SNAP-25m, the efficient decrease in donor fluorescence observed in the presence of $Ca^{2+}$, Munc18-1 and Munc13-1 $C_1C_2BMUNC_2C$ (*Figure 3A*) suggests that these factors prevent disassembly of trans-SNARE complexes in addition to mediating NSF-αSNAP-resistant assembly. Note however that we cannot completely rule out the possibility that, instead of physically preventing disassembly, Munc18-1 and Munc13-1 $C_1C_2BMUNC_2C$ mediate fast re-assembly of trans-SNARE complexes after they are disassembled. For simplicity, below we use terms like 'prevent' or 'protect against disassembly' to reflect the observation that a particular factor(s) increases the amount of assembled trans-SNARE complexes observed in the presence of NSF-αSNAP, but it is important to keep in mind both possible interpretations (see discussion).

Munc18-1 and Munc13-1 $C_1C_2BMUNC_2C$ mediate some $Ca^{2+}$-independent assembly of trans-SNARE complexes (*Figure 3A*), suggesting that these proteins also protect against disassembly in the absence of $Ca^{2+}$. However, it is unclear whether such protection is as efficient as that occurring in the presence of $Ca^{2+}$ because of the low level of $Ca^{2+}$-independent assembly observed in the time scale of these assays. To overcome this problem, we performed additional assays where $Ca^{2+}$ was added early to promote efficient trans-SNARE complex assembly, and EGTA was added afterwards to test whether, in the absence of $Ca^{2+}$, Munc18-1 and Munc13-1 $C_1C_2BMUNC_2C$ could keep the trans-SNARE complexes assembled. EGTA caused some recovery of the donor fluorescence intensity, but the recovery leveled off with time, and the later EGTA was added, the lower was the donor fluorescence at the latest time point, showing that we did not reach equilibrium in these assays (*Figure 3C*). Nevertheless, the observation that the donor fluorescence intensity at the end is markedly lower than the initial intensity shows that a population of trans-SNARE complexes remained assembled, suggesting that Munc18-1 and Munc13-1 indeed protect trans-SNARE complexes from disassembly by NSF-αSNAP in the absence of $Ca^{2+}$. Using a similar approach, we tested

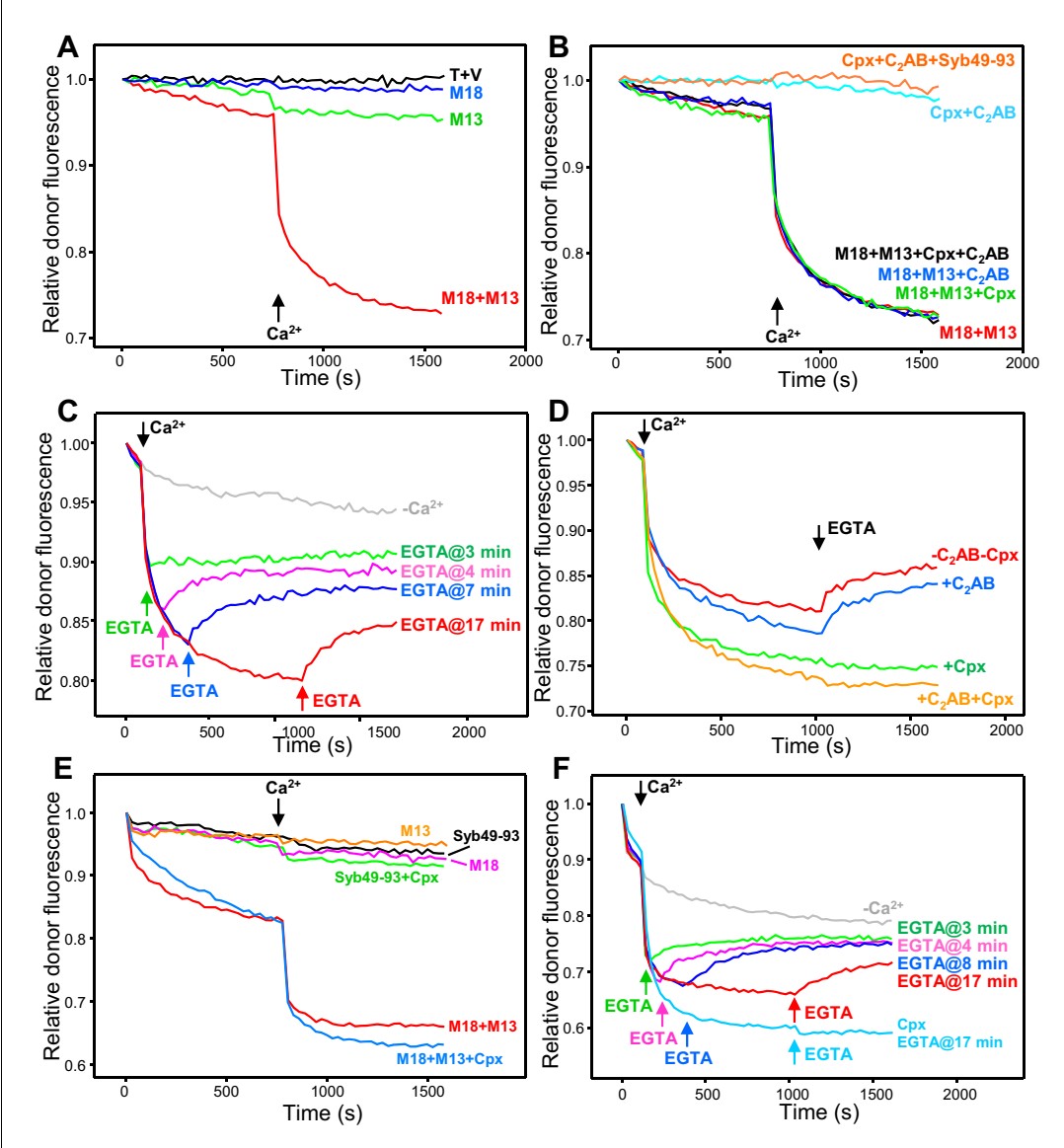

**Figure 3.** Influence of Munc18-1, Munc13-1 $C_1C_2BMUNC_2C$, complexin-1 and synaptotagmin-1 on trans-SNARE complex assembly-disassembly in the presence of NSF-αSNAP. (**A,B**) Kinetic assays monitoring trans-SNARE complex assembly between V- and T-liposomes (1:4 ratio) in the presence of NSF-αSNAP from the decrease in the donor fluorescence emission intensity. The experiments were performed in the absence of other proteins (T + V) or in the presence of different combinations of Munc18-1 (M18), Munc13-1 $C_1C_2BMUNC_2C$ (M13), complexin-1 (Cpx), synaptotagmin-1 $C_2AB$ and Syb49-93, as indicated by the colors. Experiments were started in 100 μM EGTA and $Ca^{2+}$ (600 μM) was added after 750 s. (**C**) Analogous kinetic assays performed in the presence of Munc18-1, Munc13-1 $C_1C_2BMUNC_2C$, NSF-αSNAP and 100 μM EGTA, but adding 240 μM $Ca^{2+}$ at 2 min to stimulate trans-SNARE complex assembly and adding 500 μM EGTA at different times to chelate the $Ca^{2+}$ and interrogate whether there is trans-SNARE complex disassembly. An experiment that was also started in 100 μM EGTA but without addition of $Ca^{2+}$ or EGTA at later times (gray trace) is shown for comparison. (**D**) Experiments analogous to those of (**C**), with addition of 240 μM $Ca^{2+}$ at 2 min and 500 μM EGTA at 17 min, performed in the absence or presence of complexin-1 and/or synaptotagmin-1 $C_2AB$. (**E**) Kinetic assays monitoring trans-SNARE complex assembly between VSyt1- and T-liposomes (1:4 ratio) in the presence of NSF-αSNAP and different combinations of Munc18-1, Munc13-1 $C_1C_2BMUNC_2C$, complexin-1 and Syb49-93, as indicated by the colors. Experiments were started in 100 μM EGTA and $Ca^{2+}$ (600 μM) was added after 750 s. (**F**) Kinetic assays analogous to those of (**E**) performed in the presence of Munc18-1, Munc13-1 $C_1C_2BMUNC_2C$, NSF-αSNAP and 100 μM EGTA, but adding 240 μM $Ca^{2+}$ at 2 min to stimulate trans-SNARE complex assembly and adding 500 μM EGTA at different times to chelate the $Ca^{2+}$ and interrogate whether there is trans-SNARE complex disassembly. An experiment that was also started in 100 μM EGTA but without addition of $Ca^{2+}$ or EGTA at later times (gray trace) is shown for comparison. The light blue trace shows an additional experiment started in 100 μM EGTA in the presence of complexin-1, with addition of 240 μM $Ca^{2+}$ at 2 min and 500 μM EGTA at 17 min. All experiments were performed in the presence of $Mg^{2+}$ and ATP. For all traces shown in (**A–F**), fluorescence emission intensities were normalized with the intensity observed in the first point and corrected for the dilution caused by the addition of reagents.

*Figure 3 continued on next page*

*Figure 3 continued*

DOI: https://doi.org/10.7554/eLife.38880.008

The following figure supplements are available for figure 3:

**Figure supplement 1.** Complexin-1 increases the efficiency of $Ca^{2+}$-independent trans-SNARE complex assembly between VSyt1- and T-liposomes in the presence of NSF-αSNAP.

DOI: https://doi.org/10.7554/eLife.38880.009

**Figure supplement 2.** $Ca^{2+}$-dependent fusion between VSyt1- and T-liposomes.

DOI: https://doi.org/10.7554/eLife.38880.010

whether the synaptotagmin-1 $C_2AB$ or complexin-1 protect against the disassembly of trans-SNARE complexes observed upon addition of EGTA. Including complexin-1 completely prevented such disassembly, whereas synaptotagmin-1 $C_2AB$ had no effect (*Figure 3D*). Note that complexin-1 seemed to enhance the assembly rate in these assays, in contrast to those of *Figure 3B*; thus, it is unclear from these data whether complexin-1 assists in assembly.

The use of the soluble synaptotagmin-1 $C_2AB$ fragment allowed us to directly compare the assembly and disassembly of trans-SNARE complexes in the absence and presence of the synaptotagmin-1 $C_2$ domains with the same V-liposomes, but in vivo synaptotagmin-1 is anchored on synaptic vesicles. To investigate how membrane anchoring of synaptotagmin-1 influences trans-SNARE complex assembly-disassembly, we performed FRET experiments analogous to those described above but using liposomes that contained the same low P/L ratio of Alexa488-labeled synaptobrevin (1:10,000) and synaptotagmin-1 incorporated at a 1:1,000 P/L [comparable to physiological ratios for sybaptotagmin-1; (*Takamori et al., 2006*) (referred to below as VSyt1-liposomes). Trans-SNARE complex formation between VSyt1- and T-liposomes was again stimulated strongly by the Syb49-93 peptide (*Figure 2—figure supplement 2B*) and fluorescence spectra acquired after a long incubation with Syb49-93 revealed efficient formation of trans-SNARE complexes, while addition of NSF-α SNAP disassembled about 55% of these complexes (*Figure 2—figure supplement 4C*), similar to the results obtained with V-liposomes (*Figure 2B*).

No trans-SNARE complex assembly between VSyt1- and T-liposomes was observed in kinetic experiments performed in the presence of NSF-αSNAP together with Syb49-93, Syb49-93 plus complexin-1, Munc18-1 alone or Munc13-1 alone (*Figure 3E*). However, considerable trans-SNARE complex assembly was observed in the presence of Munc18-1 and Munc13-1 $C_1C_2BMUNC_2C$, which was strongly accelerated by $Ca^{2+}$ (*Figure 3E*, red trace). These results show again that Munc18-1 and Munc13-1 $C_1C_2BMUNC_2C$ are critical for trans-SNARE complex assembly in the presence of NSF-α SNAP, as observed in the experiments performed with V- and T-liposomes (*Figure 3A,B*). Interestingly, $Ca^{2+}$-independent assembly was more efficient with the VSyt1-liposomes than with V-liposomes, suggesting that membrane-anchored synaptotagmin-1 facilitates the NSF-αSNAP-resistant assembly mediated by Munc18-1 and Munc13-1 $C_1C_2BMUNC_2C$. To test whether, in addition, membrane anchored synaptotagmin-1 helps to protect trans-SNARE complexes once they are formed, we again performed kinetic assays where we added $Ca^{2+}$ shortly after mixing the VSyt1- and T-liposomes, and EGTA was added afterwards at different time points. We again observed partial recovery of the donor fluorescence intensity upon EGTA addition (*Figure 3F*), but the overall amount of trans-SNARE complexes that remained assembled was higher than in the experiments performed with V- and T-liposomes (*Figure 3C*). These results indicate that membrane-anchored synaptotagmin-1 may help to protect trans-SNARE complexes against disassembly by NSF-αSNAP once they are formed.

In parallel experiments including complexin-1, no donor fluorescence recovery was observed when EGTA was added, showing again that complexin-1 protects against disassembly, and the overall efficiency of assembly was higher (*Figure 3F*). It is also worth noting that, in our standard assembly assays where $Ca^{2+}$ was added at 750 s, $Ca^{2+}$-independent assembly was slower in the presence of complexin-1 than in its absence, but did not appear to level off at this time, as did the reaction without complexin-1 (*Figure 3E*, red and blue curves). Indeed, at longer time periods $Ca^{2+}$-independent assembly was more efficient in the presence of complexin-1 even though it was lower in the beginning (*Figure 3—figure supplement 1*). These results suggest that, in the absence of $Ca^{2+}$, complexin-1 partially inhibits assembly of trans-SNARE complexes between VSyt1- and T-liposomes

but increases the overall assembly efficiency because it protects trans-SNARE complexes against disassembly by NSF-$\alpha$SNAP, which was further supported by additional experiments described below.

The substantial amount of $Ca^{2+}$-independent trans-SNARE complex assembly between VSyt1- and T-liposomes observed in our FRET assays in the presence of NSF-$\alpha$SNAP, Munc18-1 and Munc13-1 $C_1C_2$BMUNC$_2$C contrasts with the absence of content mixing that we commonly observed in fusion assays performed with V- or VSyt1-liposomes using a synaptobrevin-to-lipid ratio of 1:500 and incorporating WT SNAP-25 in the T-liposomes (*Figure 1C* and *Liu et al., 2016*). To verify the latter result with the same synaptobrevin density used for the trans-SNARE complex assembly assays, we performed fusion assays using VSyt1-liposomes with the same synaptobrevin-to-lipid ratio (1:10,000). We did not observe any fusion in the absence of $Ca^{2+}$ while lipid and content mixing were efficient but slow upon $Ca^{2+}$ addition (*Figure 3—figure supplement 2*), in contrast with both the substantial $Ca^{2+}$-independent trans-SNARE complex assembly and the fast $Ca^{2+}$-dependent assembly observed in our FRET assays (*Figure 3E*). These data illustrate that trans-SNARE complex assembly does not necessarily lead to membrane fusion under these conditions, as observed previously in other reconstitution assays (e.g. *Zick and Wickner, 2014*; reviewed in *Rizo, 2018*).

## Multiple factors stabilize trans-SNARE complexes against disassembly by NSF-$\alpha$SNAP

The kinetic assays of *Figure 3* show how different factors influence trans-SNARE complex assembly in the presence of NSF-$\alpha$SNAP and provide some information on which of these factors protect trans-SNARE complexes against disassembly. However, since only a fraction of trans-SNARE complexes formed by SNAREs alone are disassembled by NSF-$\alpha$SNAP (*Figure 2B*), it is plausible that the absence of disassembly during our kinetic assays arises from formation of NSF-$\alpha$SNAP-resistant trans-SNARE complexes, rather than because the various proteins actively protect against disassembly. To gain further insights into whether Munc18-1, Munc13-1, synaptotagmin-1 and complexin-1 can prevent disassembly of trans-SNARE complexes by NSF-$\alpha$SNAP, we performed similar kinetic assays where trans-SNARE complex assembly between V- and T-liposomes was monitored by FRET in the presence of various combinations of Munc18-1, Munc13-1 $C_1C_2$BMUNC$_2$C, synaptotagmin-1 $C_2$AB and complexin-1, but adding NSF-$\alpha$SNAP at the end of the reaction, rather than the beginning. For experiments with complexin-1 and synaptotagmin-1 $C_2$AB, we included the Syb49-93 peptide to facilitate trans-SNARE complex assembly, but the peptide was not included for experiments with Munc18-1 and Munc13-1 $C_1C_2$BMUNC$_2$C because these proteins presumably can overcome at least in part the inhibition arising from formation of 2:1 syntaxin-1-SNAP-25 heterodimers (*Ma et al., 2011*; *Ma et al., 2013*). We note that we did not attempt to monitor the kinetics of disassembly upon addition of NSF-$\alpha$SNAP, because disassembly generally occurred rapidly while reagents were added to multiple parallel reactions (see Materials and methods).

The rate of trans-SNARE complex assembly was similar in assays started in the absence of $Ca^{2+}$ with or without synaptotagmin-1 $C_2$AB, and the recovery of donor fluorescence upon addition of NSF-$\alpha$SNAP was also comparable (*Figure 4A*, green and black curves), showing that $C_2$AB does not alter assembly or disassembly in the absence of $Ca^{2+}$. However, assembly was dramatically accelerated by $C_2$AB in the presence of $Ca^{2+}$ (*Figure 4A*, blue curve), likely because $C_2$AB can bridge two membranes together (*Araç et al., 2006*), and addition of NSF-$\alpha$SNAP led to only a small amount of donor fluorescence recovery, suggesting that $Ca^{2+}$-bound $C_2$AB markedly protected trans-SNARE complexes against disassembly (but see below). Complexin-1 accelerated trans-SNARE complex assembly (*Figure 4A*, red curve), consistent with previous results (*Diao et al., 2013*), and appeared to partially prevent disassembly by NSF-$\alpha$SNAP. The contrast of these results with those of *Figure 3B* most likely arises because syntaxin-1-SNAP-25m heterodimers constitute the starting point for trans-SNARE complex assembly facilitated by complexin-1 and synaptotagmin-1 $C_2$AB, but this pathway is blocked in the presence of NSF-$\alpha$SNAP because they disassemble the heterodimers.

Munc13-1 $C_1C_2$BMUNC$_2$C alone or Munc18-1 alone were unable to promote trans-SNARE complex assembly, but together they did mediate trans-SNARE complex assembly that was slow in the absence of $Ca^{2+}$ and was strongly accelerated upon $Ca^{2+}$ addition (*Figure 4B*). Addition of NSF-$\alpha$SNAP consistently led to a slight further decrease in donor fluorescence intensity (*Figure 4B*, red curve), supporting the notion that Munc18-1 and Munc13-1 $C_1C_2$BMUNC$_2$C protect against trans-SNARE complex disassembly by NSF-$\alpha$SNAP, and suggesting that they in fact cooperate with NSF-$\alpha$SNAP in formation of trans-SNARE complexes. This notion is further supported by the observation

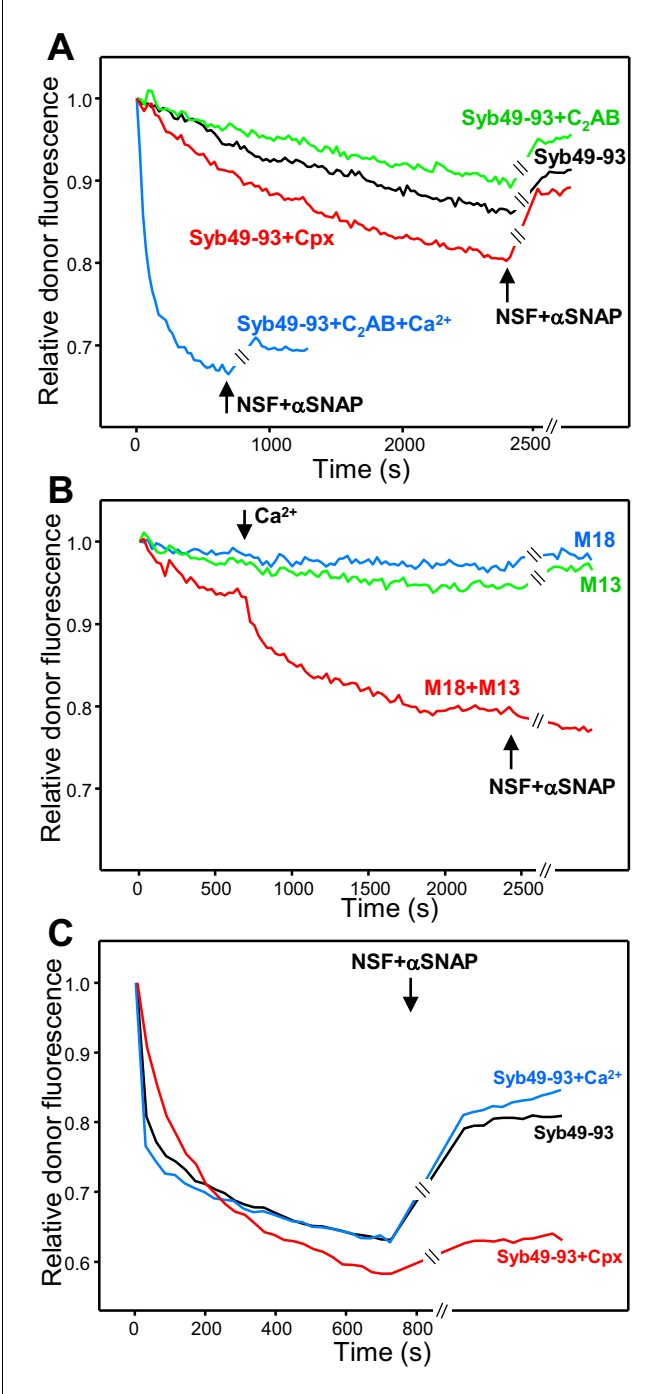

**Figure 4.** Influence of Munc18-1, Munc13-1 $C_1C_2BMUNC_2C$, complexin-1 and synaptotagmin-1 on trans-SNARE complex assembly in the absence of NSF-αSNAP and on protection against disassembly upon addition of NSF-α SNAP. (**A**) Kinetic assays monitoring trans-SNARE complex assembly upon mixing V- and T-liposomes (1:4 ratio) in the presence of Syb49-93 and disassembly upon addition of NSF-αSNAP (indicated by the arrows), from the changes in the donor fluorescence emission intensity. The experiments included Syb49-93 alone (black trace) or together with complexin-1 (Cpx) (red trace), synaptotagmin-1 $C_2AB$ (green trace) and synaptotagmin-1 plus $Ca^{2+}$ (blue trace). (**B**) Kinetic assays analogous to those of (**A**) but performed in the absence of Syb49-93 and the presence of Munc18-1 (M18), Munc13-1 $C_1C_2BMUNC_2C$ (M13) or both (blue, green and red traces, respectively). Experiments were started in 100 µM EGTA and $Ca^{2+}$ (600 µM) was added after 700 s. (**C**) Analogous kinetic assays monitoring trans-SNARE complex assembly between VSyt1- and T-liposomes (1:4 ratio) in the presence of Syb49-93 alone (black trace) or together with $Ca^{2+}$ (blue trace) or complexin-1 (Cpx) (red trace), and addition of NSF-α

*Figure 4 continued on next page*

*Figure 4 continued*

SNAP at the end (black arrow). In the experiments shown in (**A–C**), we stopped monitoring the donor fluorescence intensity to add the reagents for disassembly, and a few minutes elapsed until we started to monitor the reaction again (indicated by the double slanted bars on the traces and on the x axis). For all traces shown in (**A–C**), fluorescence emission intensities were normalized with the intensity observed in the first point and corrected for the dilution caused by the addition of reagents.

DOI: https://doi.org/10.7554/eLife.38880.011

The following figure supplement is available for figure 4:

**Figure supplement 1.** The SNARE motif of synaptobrevin does not affect the results observed upon addition of NSF-αSNAP to trans-SNARE complexes formed in the presence of Munc18-1, Munc13-1 $C_1C_2BMUNC_2C$ and $Ca^{2+}$.

DOI: https://doi.org/10.7554/eLife.38880.012

that trans-SNARE complex assembly was more efficient in the experiments performed with Munc18-1 and Munc13-1 $C_1C_2BMUNC_2C$ when NSF-αSNAP were present from the beginning (*Figure 3A*, red curve). These results most likely arise because, in the former experiments, Munc18-1 must displace the SNAP-25m bound to syntaxin-1 in the T-liposomes to initiate the Munc18-1-closed syntaxin-1 pathway. Such displacement is slow and is accelerated when NSF-αSNAP are added from the beginning because they disassemble the syntaxin-1-SNAP-25m heterodimers, facilitating binding of Munc18-1 to closed syntaxin-1 and initiating the NSF-αSNAP-resistant pathway of trans-SNARE complex assembly.

We also performed assays where we monitored formation of trans-SNARE complexes between VSyt1- and T-liposomes, including Syb49-93 to facilitate assembly and adding NSF-αSNAP at the end to test for disassembly. We observed similar rates of assembly and similar amounts of disassembly in the absence and presence of $Ca^{2+}$ (*Figure 4C*), which indicates that synaptotagmin-1 by itself does not protect against disassembly and contrasts with the results obtained with V- and T-liposomes in the presence of synaptotagmin-1 $C_2AB$ (*Figure 4A*; see discussion). Including complexin-1 decreased the assembly rate but enhanced the overall efficiency of assembly and strongly hindered disassembly of trans-SNARE complexes by NSF-αSNAP (*Figure 4C*, red curve), in correlation with the results obtained in kinetic experiments performed with NSF-αSNAP from the beginning (*Figure 3E,F*, *Figure 3—figure supplement 1*). We did not pursue these kinetic experiments further because, although they suggested that Munc18-1, Munc13-1, synaptotagmin-1 and complexin-1 have differential abilities to protect trans-SNARE complexes against disassembly NSF-αSNAP, it is difficult to quantify these abilities from these assays because of the different extent of trans-SNARE complex assembly under the various conditions, because of a small amount of photobleaching occurring during the experiments, and because it is unclear to what extent trans-SNARE complexes that are intrinsically resistant to NSF-αSNAP were formed under the various conditions.

To overcome these problems and have a common benchmark that can give a quantitative idea of the protecting activity of the various proteins, we again followed the approach of pre-forming trans-SNARE complexes by incubation of V- and T-liposomes (1:4 ratio) in the presence of Syb49-93 for five hours, after which there are no further changes in the fluorescence spectrum [note that Syb49-93 is released from the syntaxin-1-SNAP-25 complexes upon trans-SNARE complex assembly (*Yavuz et al., 2018*) and hence should not interfere in the measurement of protection against disassembly]. Different aliquots of the same reaction mixture where then incubated with synaptotagmin-1 $C_2AB$, complexin-1, Munc18-1 and Munc13-1 $C_1C_2BMUNC_2C$ in different combinations, with or without $Ca^{2+}$ whenever $C_2AB$ and/or $C_1C_2BMUNC_2C$ were present. Fluorescence emission spectra of the resulting samples were acquired before and after addition of NSF-αSNAP to quantify the changes in FRET caused by NSF-αSNAP (*Figure 5A* and *Figure 5—figure supplement 1*). We also acquired control fluorescence spectra of separate samples where we preformed trans-SNARE complexes between V-liposomes that contained the donor probe and T-liposomes lacking the acceptor probe (referred to as V*+T), as well as analogous trans-SNARE complexes that contained the acceptor probe but not the donor probe (V + T*); both sets of liposomes were also incubated with various proteins and fluorescence spectra were acquired before and after addition of NSF-αSNAP (*Figure 5—figure supplements 2* and *3*). The control spectra showed that none of the proteins substantially affect the donor fluorescence, except for a slight but consistent increase caused by Munc13-1 $C_1C_2BMUNC_2C$, while NSF-αSNAP did cause a considerable decrease of the acceptor fluorescence

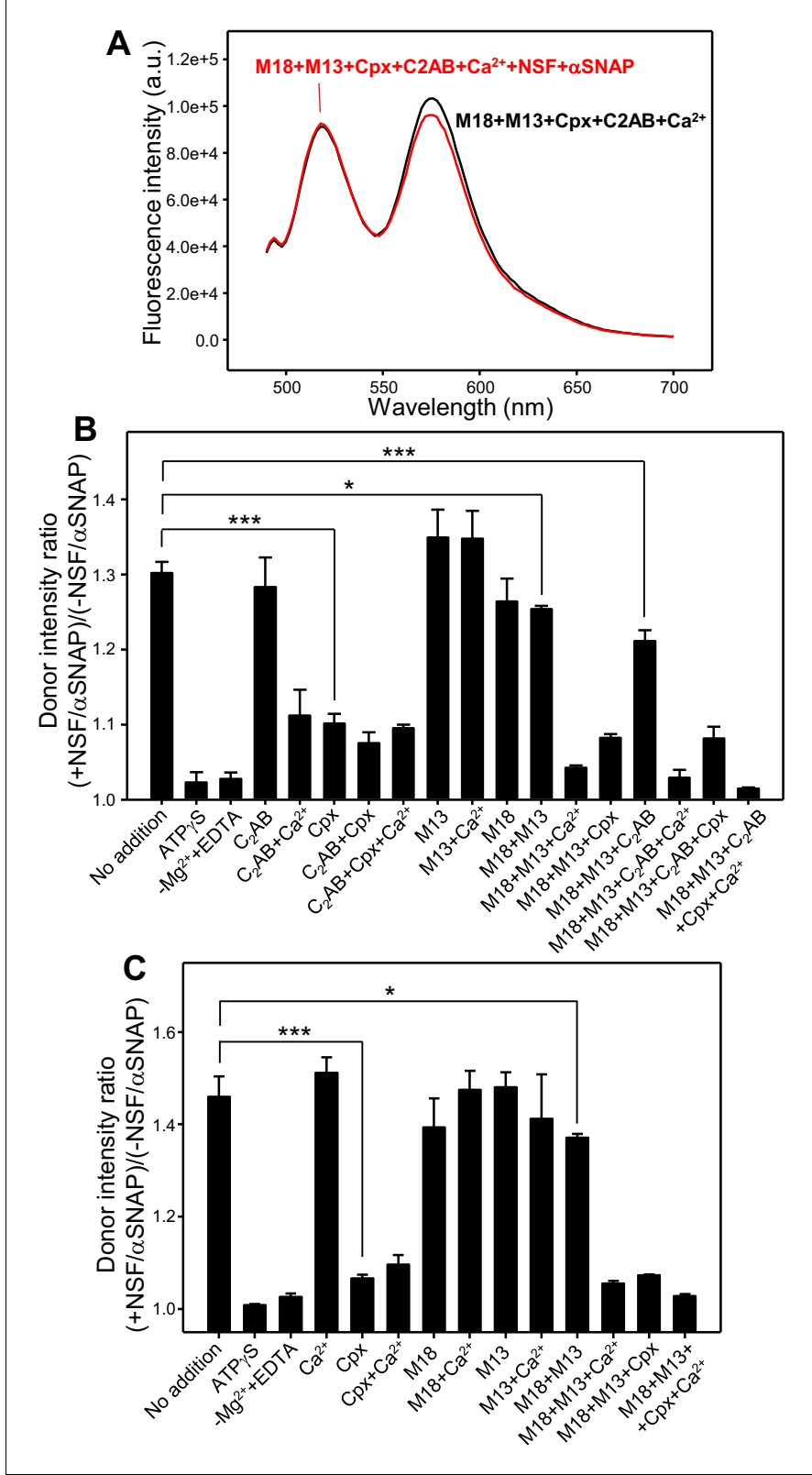

**Figure 5.** Quantitative analysis of how Munc18-1, Munc13-1 $C_1C_2BMUNC_2C$, complexin-1 and synaptotagmin-1 protect pre-formed trans-SNARE complexes against disassembly by NSF-αSNAP. (**A**) Fluorescence emission spectra of mixtures of V-liposomes containing Alexa488-synaptobrevin and T-liposomes containing TMR-syntaxin-1-SNAP-25m (1:4 V- to T-liposome ratio) that were incubated for five hours with Syb49-93; Munc18-1 (M18),
*Figure 5 continued on next page*

*Figure 5 continued*

Munc13-1 $C_1C_2BMUNC_2C$ (M13), complexin-1 (Cpx), synaptotagmin-1 $C_2AB$ ($C_2AB$) and $Ca^{2+}$ were then added and, after an additional incubation for five minutes, spectra were acquired before (black trace) or after (red trace) addition of NSF-αSNAP. (B) Bar diagram illustrating the ability of Munc18-1, Munc13-1 $C_1C_2BMUNC_2C$, complexin-1, synaptotagmin-1 $C_2AB$ and $Ca^{2+}$ to protect pre-formed trans-SNARE complexes against disassembly by NSF-αSNAP. As in (A), V- and T-liposomes were incubated for five hours with Syb49-93 to preform trans-SNARE complexes and then they were incubated for five minutes with different combinations of Munc18-1, Munc13-1 $C_1C_2BMUNC_2C$, complexin-1, synaptotagmin-1 $C_2AB$ and $Ca^{2+}$. Fluorescence emission spectra were acquired before and after addition of NSF-αSNAP and the ratio *r* between the donor fluorescence intensities at 518 nm measured after and before NSF-αSNAP addition was calculated. Representative examples of the spectra acquired under different conditions are shown in *Figure 5—figure supplement 1*. (C) Bar diagram illustrating the ability of Munc18-1, Munc13-1 $C_1C_2BMUNC_2C$, complexin-1 and $Ca^{2+}$ to protect pre-formed trans-SNARE complexes between VSyt1- and T-liposomes against disassembly by NSF-αSNAP. Similar to (B), VSyt1- and T-liposomes were incubated with Syb49-93 (but for 24 hr at 4°C) to preform trans-SNARE complexes, and then they were incubated for five minutes with different combinations of Munc18-1, Munc13-1 $C_1C_2BMUNC_2C$, complexin-1 and $Ca^{2+}$. Fluorescence emission spectra were acquired before and after addition of NSF-αSNAP and the ratio *r* between the donor fluorescence intensities at 518 nm measured after and before NSF-αSNAP addition was calculated. Representative examples of the spectra acquired under different conditions are shown in *Figure 5—figure supplement 4*. In (B,C), 'No additions' indicates experiments where none of these factors were included before addition of NSF-αSNAP. Control experiments with no additions and replacing ATP with ATPγS or replacing $Mg^{2+}$ with EDTA were also performed. All experiments were performed in triplicate. Values indicate means ±standard deviations. A few examples of statistical significance are indicated to illustrate which differences among the *r* values obtained under different conditions are meaningful. Statistical significance and P values were determined by one-way analysis of variance (ANOVA) with Holm-Sidak test (*p<0.05; ***p<0.001).

DOI: https://doi.org/10.7554/eLife.38880.013

The following figure supplements are available for figure 5:

**Figure supplement 1.** Representative fluorescence emission spectra used in the experiments of *Figure 5B* to obtain a quantitative measurement of how Munc18-1, Munc13-1, complexin-1, synaptotagmin-1 and $Ca^{2+}$ in different combinations protect pre-formed trans-SNARE complexes against disassembly by NSF-αSNAP.

DOI: https://doi.org/10.7554/eLife.38880.014

**Figure supplement 2.** Control spectra acquired to assess the effects of various factors on the fluorescence emission spectra of V-liposomes incorporated into trans-SNARE complexes in the absence of FRET.

DOI: https://doi.org/10.7554/eLife.38880.015

**Figure supplement 3.** Control spectra acquired to assess the effects of various factors on the fluorescence emission spectra of T-liposomes incorporated into trans-SNARE complexes in the absence of FRET.

DOI: https://doi.org/10.7554/eLife.38880.016

**Figure supplement 4.** Representative fluorescence emission spectra used in the experiments of *Figure 5C* to obtain a quantitative measurement of how Munc18-1, Munc13-1, complexin-1 and $Ca^{2+}$ in different combinations protect pre-formed trans-SNARE complexes between VSyt1- and T-liposomes against disassembly by NSF-α SNAP.

DOI: https://doi.org/10.7554/eLife.38880.017

**Figure supplement 5.** Complexin-1 concentration dependence of protection of trans-SNARE complexes against disassembly by NSF-αSNAP.

DOI: https://doi.org/10.7554/eLife.38880.018

in the V + T* controls that was prevented by Munc18-1. Hence, we focused on the donor fluorescence to quantitate the protection against disassembly.

The fluorescence spectra obtained after incubation of the preformed trans-SNARE complexes with different combinations of Munc18-1, Munc13-1 $C_1C_2BMUNC_2C$, synaptotagmin-1 $C_2AB$ and complexin-1 before addition of NSF-αSNAP were very similar for all samples, indicating that the amount of trans-SNARE complexes was not affected by the incubations. However, substantial differences were observed in the donor emission intensities in the spectra obtained after addition of NSF-αSNAP (*Figure 5—figure supplement 1*), indicating different extents of SNARE complex disassembly. To derive a quantitative idea of how much the different combinations of proteins protect against disassembly, we calculated the ratio *r* between the donor fluorescence intensity after adding NSF-α SNAP and that before addition of NSF-αSNAP. This ratio was 1.30 for control experiments with no additions before disassembly with NSF-αSNAP (*Figure 5B*). This value was variable in experiments

performed with different liposome preparations and depended on the extent of trans-SNARE complex assembly achieved, but the relative changes in $r$ values obtained in the presence of different factors were comparable for the different preparations.

The $r$ values measured showed that $Ca^{2+}$-free synaptotagmin-1 $C_2AB$ provided no protection but $Ca^{2+}$-bound $C_2AB$ prevented disassembly considerably. Complexin-1 afforded similar protection as $Ca^{2+}$-bound $C_2AB$. In experiments with Munc13-1 $C_1C_2BMUNC_2C$ alone in the absence or presence of $Ca^{2+}$, $r$ was slightly larger than that observed in the control with no additions, which can be attributed to the slight increase in donor fluorescence caused by Munc13-1 $C_1C_2BMUNC_2C$ on the V*+T control (*Figure 5—figure supplement 2C*) and shows that there is no protection against disassembly under these conditions. The $r$ value observed with Munc18-1 alone was slightly smaller than 1.3; although the difference with respect to the controls with no additions was not statistically significant, there was a significant difference between the (smaller) $r$ value observed in experiments with Munc18-1 and Munc13-1 $C_1C_2BMUNC_2C$ in the absence of $Ca^{2+}$ and the control with no additions. A dramatic decrease in $r$ was observed when $Ca^{2+}$ was included with Munc18-1 and Munc13-1 $C_1C_2BMUNC_2C$. Adding complexin-1 or of $C_2AB$ together with Munc18-1 and Munc13-1 $C_1C_2BMUNC_2C$, with or without $Ca^{2+}$, also decreased the corresponding $r$ values, and the smallest $r$ was observed in experiments that included all these components (*Figure 5B*), showing almost complete protection under these conditions (*Figure 5A*). Overall, these results support the notion that Munc18-1 and Munc13-1 $C_1C_2BMUNC_2C$ can protect trans-SNARE complexes against disassembly to a moderate extent in the absence of $Ca^{2+}$ and that such protection is increased by $Ca^{2+}$, in correlation with the results of the kinetic assays (*Figure 3*). In addition, these data also indicate that complexin-1 also protects against disassembly and that $Ca^{2+}$-free synaptotagmin-1 $C_2AB$ alone does not prevent disassembly but can help to protect against disassembly.

To investigate how protection of trans-SNARE complexes is influenced by membrane-anchored synaptotagmin-1, we performed additional experiments where we preformed trans-SNARE complexes between VSyt1- and T-liposomes in the presence of Syb49-93, and we incubated the resulting samples with different combinations of Munc18-1, Munc13-1 $C_1C_2BMUNC_2C$, complexin-1 and $Ca^{2+}$ before adding NSF-αSNAP to test for disassembly. Munc18-1, Munc13-1 $C_1C_2BMUNC_2C$, complexin-1 and $Ca^{2+}$ did not significantly alter the fluorescence spectra acquired before addition of NSF-αSNAP, but markedly affected the spectra obtained after such addition (black and red curves, respectively, in the different panels of *Figure 5—figure supplement 4*). The ratio $r$ between the donor fluorescence emission intensities observed after and before addition of NSF-αSNAP without other proteins was 1.46 and, surprisingly, addition of $Ca^{2+}$ did not lead to protection against disassembly (*Figure 5C*), which contrasts with the protection provided by $Ca^{2+}$-bound synaptotagmin-1 $C_2AB$ in experiments with V-liposomes (*Figure 5B*) and suggests that the latter result might arise from excessive accumulation of $C_2AB$ molecules at the membrane-membrane interface (*Araç et al., 2006*). Munc18-1 alone again appeared to have a tendency to prevent disassembly, compared to the control with no additions, but the difference was not statistically significant, and Munc13-1 $C_1C_2BMUNC_2C$ alone provided no protection. Together, Munc18-1 and Munc13-1 $C_1C_2BMUNC_2C$ did provide moderate protection in the absence of $Ca^{2+}$ and strong protection in its presence. Interestingly, complexin-1 alone afforded robust protection against disassembly (*Figure 5C*) that appeared to be stronger than that observed with V- and T-liposomes (*Figure 5B*), suggesting that membrane-anchored synaptotagmin-1 can cooperate with complexin-1 in protecting trans-SNARE complexes against disassembly. Maximal protection of the trans-SNARE complexes between VSyt1- and T-liposomes against disassembly by NSF-αSNAP was again observed when all components (Munc18-1, Munc13-1 $C_1C_2BMUNC_2C$, complexin-1 and $Ca^{2+}$) were included (*Figure 5C*).

The levels of protection afforded by different proteins in these experiments need to be examined with caution, as they are expected to depend on their concentrations. This is exemplified by the lower protection provided by complexin-1 as we decreased its concentration from 2 μM (used in our standard assays, *Figure 5B,C*) to 0.2 μM (*Figure 5—figure supplement 5*). Note also that we kept the concentration of Munc13-1 $C_1C_2BMUNC_2C$ at 0.3 μM because higher concentrations sometimes led to sample precipitation, but in vivo the local concentrations of Munc13-1 at the active zone may be highly increased due to binding to RIMs (see discussion). Nevertheless, the overall results presented in *Figures 3–5* provide strong evidence that Munc18-1, Munc13-1, complexin-1 and likely synaptotagmin-1 contribute to protect trans-SNARE complexes against disassembly by NSF-αSNAP.

## Disassembly of cis-SNARE complexes

To further investigate the functional interplay between NSF, αSNAP, Munc18-1, Munc13-1, complexin-1 and synaptotagmin-1 in the SNARE complex assembly-disassembly cycle, we performed kinetic assays where we analyzed the assembly and disassembly of cis-SNARE complexes mixing V-liposomes containing Alexa488-synaptobrevin with SNAP-25m and a soluble fragment spanning the cytoplasmic region of syntaxin-1 (residues 2–253) labeled with TMR at residue 186. Cis-SNARE complex assembly was efficient in the presence of Syb49-93 but was abolished if NSF-αSNAP were

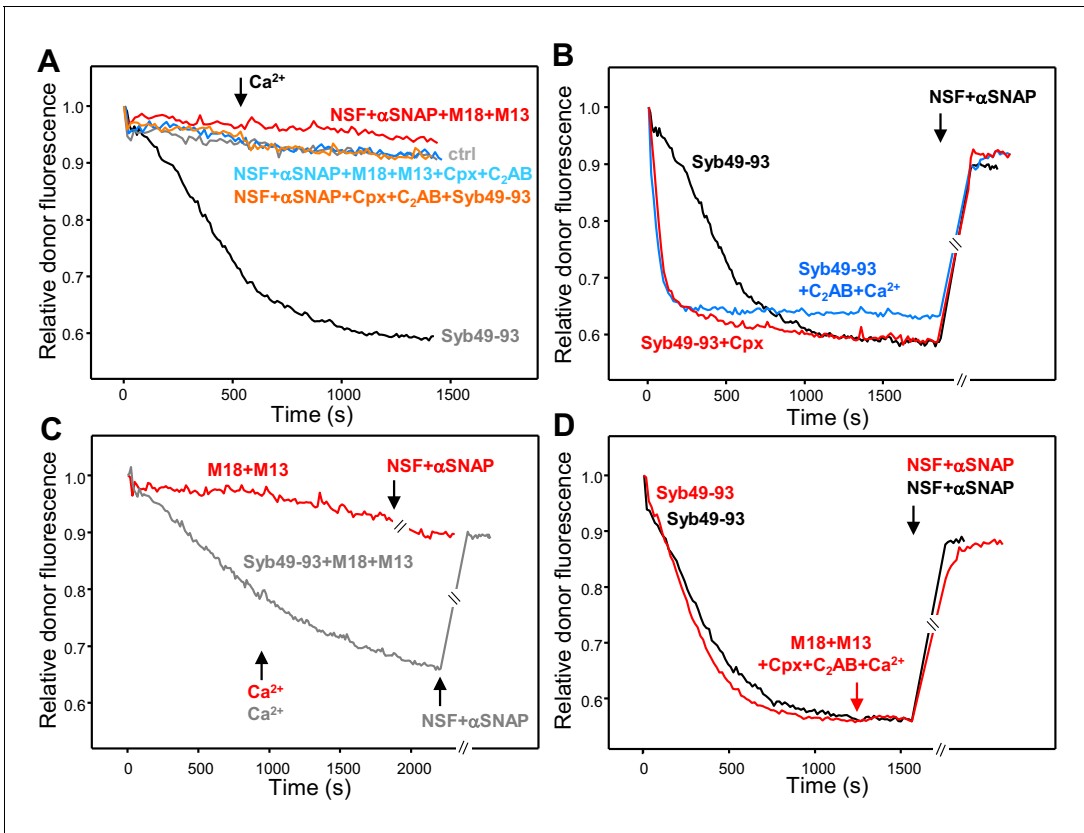

**Figure 6.** Munc18-1, Munc13-1 $C_1C_2BMUNC_2C$, complexin-1 and synaptotagmin-1 $C_2AB$ do not decrease the overall amount of cis-SNARE complex disassembly caused by NSF-αSNAP. (**A**) Kinetic assays monitoring changes in the donor fluorescence emission intensity due to cis-SNARE complex formation upon mixing V-liposomes containing Alexa488-synaptobrevin with an excess of TMR-labeled syntaxin-1 (2–253) and SNAP-25m in the presence of NSF-αSNAP with no additions (ctrl) (light gray trace) or with different combinations of Munc18-1 (M18), Munc13-1 $C_1C_2BMUNC_2C$ (M13), complexin-1 (Cpx) and synaptotagmin-1 $C_2AB$ as indicated. $Ca^{2+}$ was added at 550 s. For comparison purposes, the dark gray trace shows a cis-SNARE complex assembly reaction performed in the presence of Syb49-93 and absence of NSF-αSNAP. (**B**) Kinetic assays of cis-SNARE complex assembly analogous to those of (**A**), but performed in the absence of NSF-αSNAP and the presence of Syb49-93 alone (black trace) or together with complexin-1 (Cpx) (red trace) or synaptotagmin-1 $C_2AB$ plus $Ca^{2+}$ (blue trace). NSF-αSNAP were added when the reactions reached a plateau (black arrow) to monitor cis-SNARE complex disassembly. (**C**) Kinetic assays analogous to those in (**B**), but in the presence of Munc18-1 (M18) and Munc13-1 $C_1C_2BMUNC_2C$ (M13) without (red trace) or with (dark gray trace) Syb49-93. $Ca^{2+}$ was added after 950 s. (**D**) Kinetic assays where cis-SNARE complex formation was initially catalyzed by Syb49-93 and, after reaching a plateau, Munc18-1, Munc13-1 $C_1C_2BMUNC_2C$, complexin-1, synaptotagmin-1 $C_2AB$ and $Ca^{2+}$ were added (red arrow); after five minutes, NSF-αSNAP were added to test for disassembly (red trace). The black trace shows a control experiment where the four proteins were not included before adding NSF-αSNAP. In the experiments shown in (**C–D**), we stopped monitoring the donor fluorescence intensity to add the reagents for disassembly, and a few minutes elapsed until we started to monitor the reaction again (indicated by the double slanted bars on the traces and on the x axis). For all traces of (**A–D**), fluorescence emission intensities were normalized with the intensity observed in the first point and corrected for the dilution caused by the addition of reagents.

DOI: https://doi.org/10.7554/eLife.38880.019

The following figure supplement is available for figure 6:

**Figure supplement 1.** Munc18-1, Munc13-1 $C_1C_2BMUNC_2C$, complexin-1 and synaptotagmin-1 $C_2AB$ do not protect cis-SNARE complexes against disassembly by NSF-αSNAP.

DOI: https://doi.org/10.7554/eLife.38880.020

included from the beginning (*Figure 6A*, dark and light gray curves, respectively). Munc18-1 plus Munc13-1 $C_1C_2$BMUNC$_2$C or complexin-1 plus synaptotagmin-1 $C_2$AB, or the four proteins together, were unable to support cis-SNARE complex formation in the presence of NSF-αSNAP even upon addition of $Ca^{2+}$ (*Figure 6A*). These results are in stark contrast to the efficient formation of trans-SNARE complexes observed in the presence of NSF-αSNAP when Munc18-1 and Munc13-1 $C_1C_2$BMUNC$_2$C were included (*Figure 3A*).

In experiments performed initially without NSF-αSNAP, cis-SNARE complex assembly was strongly stimulated by complexin-1 or by $Ca^{2+}$-bound synaptotagmin-1 $C_2$AB, but most of the donor fluorescence was recovered upon addition of NSF-αSNAP at the end of the reaction due to disassembly of the cis-SNARE complexes (*Figure 6B*). Munc18-1 and the Munc13-1 $C_1C_2$BMUNC$_2$C fragment were unable to support cis-SNARE complex assembly even after addition of $Ca^{2+}$ (*Figure 6C*, red curve), and they partially inhibited cis-SNARE complex assembly catalyzed by Syb49-93, without protecting against disassembly upon addition of NSF-αSNAP (*Figure 6C*, gray curve). We also performed additional experiments where we preformed cis-SNARE complexes in the presence of Syb49-93 and tested whether incubation of these complexes with Munc18-1, Munc13-1, synaptotagmin-1 $C_2$AB, complexin-1 and $Ca^{2+}$ for five minutes protected against disassembly by NSF-αSNAP, but disassembly was as efficient as a control experiment where the four proteins were not added (*Figure 6D*). We note that these results are not necessarily inconsistent with the finding that complexin-1 slows down the kinetics of cis-SNARE complex disassembly by NSF-αSNAP (*Choi et al., 2018*; *Winter et al., 2009*), as we did not attempt to measure the kinetics of disassembly in our experiments (see Materials and methods). Nevertheless, to test for potential effects on the overall extent of disassembly arising from different relative concentrations of complexin-1 versus αSNAP, or perhaps from the mutation in SNAP-25m, we performed additional experiments where we incubated pre-formed cis-SNARE complexes with different concentrations of complexin-1 or we replaced SNAP-25m with WT SNAP-25. We observed comparable, nearly complete levels of disassembly in all of these experiments (*Figure 6—figure supplement 1*). Overall, the contrast of the results obtained with cis-SNARE complexes with those observed with trans-SNARE complexes provides a dramatic demonstration of how the apposition of two membranes tilts the balance in favor of SNARE complex assembly, whereas disassembly dominates on a single membrane.

## Discussion

Extensive research has yielded a wealth of information on the mechanism of neurotransmitter release, including the notions that assembly of the trans-SNARE complex four-helix bundle between the synaptic vesicle and plasma membranes is crucial for membrane fusion, that NSF-αSNAP disassemble cis-SNARE complexes after fusion to recycle the SNAREs, and that priming of synaptic vesicles to a readily releasable state involves formation of partially assembled trans-SNARE complexes, which is organized by Munc18-1 and Munc13-1 in an NSF-αSNAP-resistant manner. However, the nature of the primed state of synaptic vesicles remained enigmatic and recent reports indicating that NSF-SNAPs also disassemble trans-SNARE complexes (*Yavuz et al., 2018*) and can de-prime synaptic vesicles (*He et al., 2017*) raised the question of how trans-SNARE complexes are protected to prevent vesicle de-priming. More generally, it was unclear how the functions of Munc18-1 and Munc13-1, as well as those of other proteins that have been implicated in vesicle priming such as synaptotagmin-1 and complexin-1, are related to roles in promoting trans-SNARE complex assembly and/or in preventing their disassembly by NSF-αSNAP. The results presented here now show that Munc18-1 and Munc13-1 are crucial to form trans-SNARE complexes in the presence of NSF-αSNAP, as expected, and help to maintain trans-SNARE complexes assembled. Complexin-1 does not appear to play a role in NSF-αSNAP-resistant trans-SNARE complex assembly, but protects against disassembly, while synaptotagmin-1 may play a role in both assembly and protection. These results raise the possibility that Munc18-1, Munc13-1, synaptotagmin-1 and complexin-1 form macromolecular assemblies with trans-SNARE complexes that constitute the core of the primed state of synaptic vesicles.

Our FRET data showing that trans-SNARE complexes can be disassembled by NSF-αSNAP agree with recent results obtained by *Yavuz et al. (2018)* using a similar approach, and with earlier studies of yeast vacuolar fusion showing that Sec18-Sec17 disassemble trans-SNARE complexes (*Xu et al., 2010*). However, our FRET assays and those of *Yavuz et al. (2018)* also show that a substantial

fraction of trans-SNARE complexes is resistant to disassembly by NSF-αSNAP, which might explain the finding that NSF-αSNAP inhibited lipid mixing between synaptobrevin- and syntaxin-1-SNAP-25 liposomes if added from the beginning but not if added after the liposomes were pre-incubated at low temperature (*Weber et al., 2000*). NSF-αSNAP resistant, tightly docked liposomes were attributed to the formation of large, flat interfaces between the liposomes (*Yavuz et al., 2018*). Our cryo-EM images also revealed extended interfaces between liposomes but the interfaces were generally smaller (*Figure 2—figure supplement 3*), perhaps because we used a much lower synaptobrevin-to-lipid ratio. It is unclear whether such extended interfaces are physiologically relevant, as inclusion of other key components of the release machinery favors the formation of point contacts between liposomes over extended interfaces (*Gipson et al., 2017*). These observations emphasize the difficulty of reconstituting with a few components the steps that lead to synaptic vesicle fusion, particularly the formation of the primed state, because of the metastable, transient nature of this state and because off-pathway, kinetically trapped states can be formed in the absence of some components that are important for vesicle priming (e.g. RIM and CAPS in our assays, see (*Rizo and Südhof, 2012*). We speculate that the population of trans-SNARE complexes that can be disassembled by NSF-αSNAP in our assays is more closely related to the partially assembled trans-SNARE complexes present in primed synaptic vesicles. This proposal is supported by electrophysiological studies showing that readily-releasable vesicles can be de-primed and that de-priming is prevented by N-ethylmaleimide, an agent that inactivates NSF (*He et al., 2017*). Although N-ethylmaleimide could potentially alter other proteins in vivo, the correlation with the finding that trans-SNARE complexes can be disassembled by NSF-αSNAP in vitro strongly supports the notion that de-priming is mediated by NSF. Since NSF-SNAPs can also disassemble syntaxin-1-SNAP-25 heterodimers (*Hayashi et al., 1995*), there is little doubt that the cytoplasm provides an environment that favors SNARE complex disassembly in general, and hence that trans-SNARE complexes need to be protected to maintain vesicles primed.

The decreases in the RRP of primed vesicles observed in mice lacking Munc18-1, Munc13-1, complexins or synaptotagmin-1/7 (*Bacaj et al., 2015*; *Chang et al., 2018*; *Rosenmund et al., 2002*; *Verhage et al., 2000*; *Xue et al., 2010*; *Yang et al., 2010*) could arise because they mediate vesicle priming and/or because they protect against de-priming. With the underlying hypothesis that trans-SNARE complex assembly in our in vitro assays recapitulates at least to some extent the process of vesicle priming, we used different types of assays to dissect the contributions of Munc18-1, Munc13-1, synaptotagmin-1 and complexin-1 to assembling trans-SNARE complexes in the presence of NSF-αSNAP and to protecting these complexes against disassembly once they are formed. Our assays that included NSF-αSNAP from the beginning clearly show that Munc18-1 and Munc13-1 $C_1C_2$BMUNC$_2$C are essential to assemble trans-SNARE complexes in the presence of NSF-αSNAP (*Figure 3A,B,E*), as expected from the results of our previous liposome fusion assays (*Liu et al., 2016*; *Ma et al., 2013*). The progressive formation of trans-SNARE complexes observed in these assays suggests that Munc18-1 and Munc13-1 $C_1C_2$BMUNC$_2$C prevent their disassembly, in addition to mediating assembly, but we could not rule out that the assembled trans-SNARE complexes are NSF-αSNAP resistant and Munc18-1 and/or Munc13-1 $C_1C_2$BMUNC$_2$C become dispensable after assembly, particularly in the absence of $Ca^{2+}$. The experiments where we added EGTA after allowing efficient $Ca^{2+}$-dependent assembly show that at least a population of the trans-SNARE complexes formed could be disassembled by NSF-αSNAP, but a substantial amount of complexes remained assembled even after addition of EGTA (*Figure 3C,F*). These data suggest that Munc18-1 and Munc13-1 $C_1C_2$BMUNC$_2$C do protect trans-SNARE complexes against disassembly by NSF-αSNAP to some extent, and that $Ca^{2+}$ enhances the protective activity. This conclusion was further supported by experiments where we preformed trans-SNARE complexes in the absence of NSF-αSNAP and monitored disassembled by NSF-αSNAP in the presence of Munc18-1 and Munc13-1 $C_1C_2$BMUNC$_2$C (*Figure 5B,C*).

An alternative interpretation of these results is that Munc18-1 and Munc13-1 $C_1C_2$BMUNC$_2$C do not prevent disassembly but instead mediate fast re-assembly of trans-SNARE complexes after they are disassembled by NSF-αSNAP. Although we cannot completely rule out this possibility, multiple arguments support the notion that that Munc18-1 and Munc13-1 $C_1C_2$BMUNC$_2$C directly or indirectly hinder the disassembly reaction. First, this mechanism makes more sense from an energetic point of view, as it does not involve futile cycles of disassembly and re-assembly. Second, in kinetic assays where trans-SNARE complexes were assembled in the presence of Munc18-1, Munc13-1

$C_1C_2BMUNC_2C$ and $Ca^{2+}$, and NSF-αSNAP were added at the end (**Figure 4B**), we observed similar results if NSF-αSNAP were added together with an excess of the synaptobrevin SNARE motif (**Figure 4—figure supplement 1**). If continued disassembly and re-assembly of SNARE complexes occurred under these conditions, the gradual incorporation of the soluble synaptobrevin fragment into SNARE complexes would be expected to decrease the observed FRET, but no such decrease was observed. Third, the finding that both Munc18-1 and Munc18-2 can mediate priming but only Munc18-1 prevents de-priming by NSF in neurons (**He et al., 2017**) suggests that both isoforms can mediate trans-SNARE complex assembly but only Munc18-1 prevents disassembly. Fourth, αSNAP was reported to strongly inhibit liposome lipid mixing by binding to trans-SNARE complexes (**Park et al., 2014**). Hence, fusion might be arrested after Munc18-1 and Munc13-1 $C_1C_2BMUNC_2C$ organize trans-SNARE complex assembly unless they block αSNAP binding. And fifth, Munc18-1 and Munc13-1 exhibit weak interactions with SNARE complexes in solution that are strengthened by membranes (**Dulubova et al., 2007**; **Guan et al., 2008**; **Ma et al., 2011**; **Shen et al., 2007**; **Weninger et al., 2008**), and could compete with binding of αSNAP to the SNARE four-helix bundle. Indeed, αSNAP covers much of the surface of the SNARE four-helix bundle in the cryo-EM structure of the 20S complex formed by NSF, αSNAP and the SNAREs (**Zhao et al., 2015**) (**Figure 7A**), and therefore almost any protein that interacts with the SNARE four-helix bundle might compete with αSNAP for binding. Note also that Munc13-1 $C_1C_2BMUNC_2C$ was recently proposed to bridge the vesicle and plasma membranes (**Liu et al., 2016**), which might provide an additional mechanism to protect trans-SNARE complexes against disassembly by NSF-αSNAP by imposing steric constraints that hinder formation of the 20S complex. Moreover, $Ca^{2+}$-binding to the Munc13-1 $C_2B$ domain is expected to change the orientation of Munc13-1 $C_1C_2BMUNC_2C$ with respect to the plasma membrane, bringing the two membranes into closer proximity (**Xu et al., 2017**) and potentially increasing the steric constraints that impair 20S complex assembly. This model can explain why $Ca^{2+}$ increases

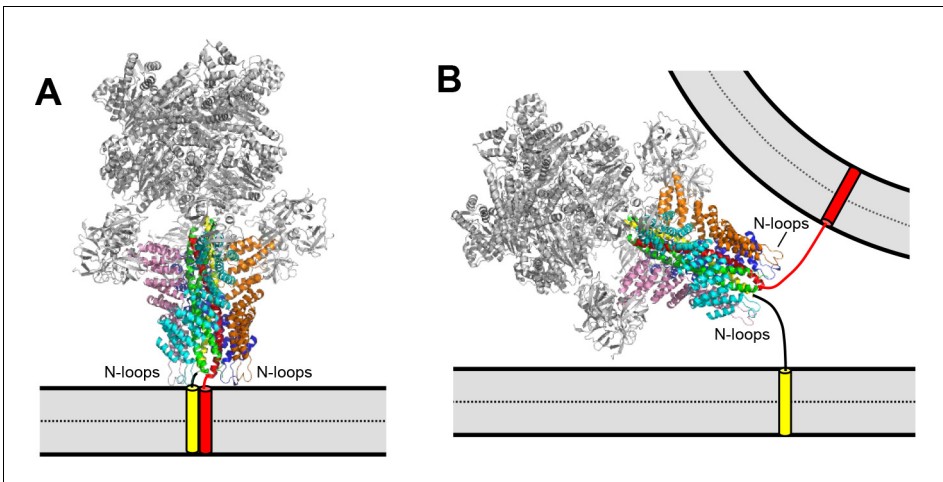

**Figure 7.** Models illustrating the different geometric constraints of cis- and trans-SNARE complex disassembly. (**A, B**) Models showing ribbon diagrams of the cryo-electron microscopy structure of the 20S complex (PDB accession code 3J96) (**Zhao et al., 2015**) assembled on a cis-SNARE complex on one membrane (**A**) or on a trans-SNARE complex between two membranes (**B**). Synaptobrevin is in red, syntaxin-1 in yellow, SNAP-25 in green, NSF in gray and the four molecules of αSNAP in cyan, orange, blue and pink. The positions of the αSNAP N-terminal hydrophobic loops (N-loops) are indicated. The orientation of the 20S complex in (**A**) was chosen to favor simultaneous interactions of the N-loops of the four αSNAP molecules with the membrane. In (**B**), the orientation of the 20S complex is arbitrary and is meant to illustrate the difficulty of simultaneous interactions of the N-loops from the four αSNAP molecules with membranes in the trans configuration. Note that, at the same time, the apposition of both membranes may enhance the affinity of Munc18-1, Munc13-1, synaptotagmin-1 and complexin-1 for SNARE complexes in the trans configuration due to simultaneous interactions with the membranes that are not possible or less favorable in the cis configuration, while the SNARE four-helix bundle is likely to be only partially assembled, which may weaken binding to αSNAP.
DOI: https://doi.org/10.7554/eLife.38880.021

the ability of Munc13-1 $C_1C_2BMUNC_2C$ (together with Munc18-1) to protect trans-SNARE complexes against disassembly by NSF-$\alpha$SNAP (*Figure 5B,C*).

Overall, the crucial nature of Munc18-1 and Munc13-1 $C_1C_2BMUNC_2C$ for trans-SNARE complex assembly provides a clear explanation for the complete abrogation of synaptic vesicle priming observed in mice lacking Munc18-1 or Munc13-1/2 (*Varoqueaux et al., 2002*; *Verhage et al., 2000*), while the finding that Munc18-1 and Munc13-1 $C_1C_2BMUNC_2C$ maintain trans-SNARE complexes assembled in the presence of NSF-$\alpha$SNAP can explain the key importance of Munc18-1 and Munc13-1 to prevent de-priming of the RRP (*He et al., 2017*). These correlations support the physiological relevance of our in vitro results. It is also worth noting that the strong $Ca^{2+}$-dependent stimulation of the ability of Munc18-1 and Munc13-1 $C_2B$ domain to mediate NSF-$\alpha$SNAP-resistant trans-SNARE complex assembly (*Figure 3A,E*) and to maintain trans-SNARE complexes assembled (*Figure 5B,C*) likely underlies at least in part the importance of $Ca^{2+}$ binding to the Munc13 $C_2B$ domain for facilitating release during repetitive stimulation, when there is a strong demand to rapidly refill the RRP to prevent its depletion (*Shin et al., 2010*).

Synaptotagmin-1 and complexin-1 are not essential to form trans-SNARE complexes in the presence of NSF-$\alpha$SNAP, but they do enhance the amount of trans-SNARE complexes formed (*Figure 3*). These findings correlate with the observation that deletion of synaptotagmin-1 or complexins leads to decreases in the RRP of vesicles (*Bacaj et al., 2015*; *Chang et al., 2018*; *Xue et al., 2010*; *Yang et al., 2010*), but not as dramatic as those observed in Munc18-1 KO and Munc13-1/2 DKO neurons. Complexin-1 appeared to inhibit in the initial states of $Ca^{2+}$-independent trans-SNARE complex assembly between VSyt1- and T-liposomes in the presence of Munc18-1, Munc13-1 $C_1C_2BMUNC_2C$ and NSF-$\alpha$SNAP, but increased the overall efficiency of assembly at longer time scales (*Figure 3E*, *Figure 3—figure supplement 1*). Moreover, complexin-1 clearly protected against disassembly by NSF-$\alpha$SNAP in analogous experiments where EGTA was added after efficient $Ca^{2+}$-dependent assembly (*Figure 3D,F*), and in experiments where NSF-$\alpha$SNAP were added after trans-SNARE complexes were pre-formed (*Figure 5B,C*). These results suggest that complexin-1 does not assist in synaptic vesicle priming but protects the RRP against de-priming. Conversely, membrane-anchored synaptotagmin-1 accelerated $Ca^{2+}$-independent trans-SNARE complex assembly (*Figure 3A,E*), suggesting a role in priming, but did not protect by itself against disassembly of pre-formed trans-SNARE complexes (*Figure 5C*). Moreover, $Ca^{2+}$-free synaptotagmin-1 $C_2AB$ did not protect pre-formed trans-SNARE complexes against disassembly and the protection provided by $Ca^{2+}$-bound $C_2AB$ may not be meaningful (see above) (*Figure 5B*). However, $C_2AB$ appeared to enhance the protection against disassembly provided by complexin-1 and by Munc18-1 plus Munc13-1 in the absence of $Ca^{2+}$ (*Figure 5B*), and membrane-anchored synaptotagmin-1 also seemed to enhance the protection of pre-formed trans-SNARE complexes afforded by complexin-1 (*Figure 5C*), as well as the levels of trans-SNARE complexes formed in the presence of NSF-$\alpha$SNAP, Munc18-1 and Munc13-1 $C_1C_2BMUNC_2C$ (*Figure 3A,C,E,F*). These results suggest that synaptotagmin-1 cooperates with Munc18-1-Munc13-1 in priming and may also help in preventing de-priming.

Mechanistically, it is not surprising that complexin-1 can hinder formation of the 20S complex, preventing disassembly of trans-SNARE complexes, as it binds to SNARE complexes with nanomolar affinity (*Pabst et al., 2002*) through a binding mode (*Chen et al., 2002*) that is incompatible with formation of the 20S complex. The various binding modes that have been observed between synaptotagmin-1 and the SNARE complex (*Brewer et al., 2015*; *Zhou et al., 2015*; *Zhou et al., 2017*) are also incompatible with $\alpha$SNAP binding. Although the affinities of these interactions are in the micromolar range, binding could be enhanced by the localization of synaptotagmin-1 on the vesicle membrane and by cooperativity with complexin-1 binding (*Zhou et al., 2017*) as well as with interactions of synaptotagmin-1 with one or two membranes (*Bai et al., 2004*; *Brewer et al., 2015*). It is also plausible that trans-SNARE complexes bound to complexin-1 and synaptotagmin-1 form macromolecular assemblies with Munc18-1 and Munc13-1, and that all underlying interactions cooperate with each other. In such an assembly, Munc18-1 and/or Munc13-1 may not interact directly with the SNAREs, but the cooperativity of the interactions and the resulting geometry might block access of $\alpha$SNAP and NSF to the SNARE four-helix bundle and thus prevent its disassembly.

It is interesting to note the dramatic effects that the membrane topology has on SNARE complex assembly and on protection against disassembly: Munc18-1 and Munc13-1 $C_1C_2BMUNC_2C$ mediate efficient formation of trans-SNARE complexes but not of cis-SNARE complexes in the presence of NSF-$\alpha$SNAP (*Figures 3A* and *6A*), and pre-formed trans-SNARE complexes remain assembled in the

presence of NSF-αSNAP when Munc18-1, Munc13-1 $C_1C_2$BMUNC$_2$C, complexin-1 and synaptotagmin-1 are included (*Figure 5B,C*), unlike cis-SNARE complexes (*Figure 6D*). These differences must arise from distinct balances among the interactions of these proteins with the SNAREs and the membranes. In the cis configuration, up to four αSNAP molecules interact with much of the surface of the SNARE four-helix bundle (*Zhao et al., 2015*) and at the same time a hydrophobic N-terminal loop from all αSNAP molecules, which is known to strongly stimulate disassembly of membrane-anchored cis-SNARE complexes (*Winter et al., 2009*), can bind simultaneously to the membrane, likely with high cooperativity (*Figure 7A*). Interactions of αSNAP with the SNAREs and the membranes are also important for disassembly of trans-SNARE complexes by NSF-αSNAP (*Figure 2D,E*), but the geometry of the system (*Figure 7B*) is expected to hinder simultaneous binding of all αSNAP molecules to membranes, and incomplete assembly of the SNARE four-helix bundle may also limit the extent of αSNAP-SNARE interactions (perhaps enhancing other interactions). Hence, NSF-αSNAP are expected to be less active in disassembling trans- than cis-SNARE complexes. Conversely, the trans-configuration favors simultaneous binding of Munc13-1 $C_1C_2$BMUNC$_2$C to the apposed membranes, which is likely key for its activity in promoting trans-SNARE complex assembly (*Liu et al., 2016*) and is impossible in the cis-configuration. The protection of trans- but not cis-SNARE complexes by complexin-1 under the conditions of our experiments may arise simply because complexin-1 binds tighter to the former than NSF-αSNAP, while the opposite is true for the latter. This model also explains that complexin-1 does hinder the speed of disassembly of cis-SNARE complexes (*Choi et al., 2018*; *Winter et al., 2009*), which we did not analyze in our experiments.

As mentioned above, the levels of protection of trans-SNARE complex against disassembly by NSF-αSNAP that we observed (*Figure 5B,C*) are also expected to depend on the experimental conditions, including protein concentrations (e.g. *Figure 5—figure supplement 5*), and hence need be examined with caution. Moreover, other proteins that were not included in this study may also influence the protection of trans-SNARE complexes directly and/or by enhancing the local concentrations of protecting factors. For instance, RIMs are intrinsic components of pre-synaptic active zones that bind to Munc13-1 (*Betz et al., 2001*; *Dulubova et al., 2005*), an interaction that is important for optimal vesicle priming (*Camacho et al., 2017*) and can dramatically increase the local concentrations of Munc13-1 at release sites. Thus, more systematic studies of how the components of the release machinery protect trans-SNARE complexes against disassembly in vitro and against de-priming of the RRP in neurons will be required to better understand the nature of the primed state of synaptic vesicles. Based on the available data, we propose that the core of this primed state is formed by a macromolecular assembly that includes trans-SNARE complexes, Munc18-1, Munc13-1, complexin-1 and synaptotagmin-1.

## Materials and methods

### Recombinant proteins

The following constructs were used for protein expression in *E. coli* BL21 (DE3) cells: Full-length rat syntaxin-1A, rat syntaxin 2–253, full-length rat SNAP-25A (C84S, C85S, C90S, C92S), full-length rat synaptobrevin, rat synaptobrevin 49–93, rat synaptotagmin-1 57–421 (C74S, C75A, C77S, C79I, C82L, C277S) (a kind gift from Thomas Sollner), rat synaptotagmin-1 $C_2$AB (131–421 C277A), full-length rat complexin, full-length Chinese hamster NSF (a kind gift from Minglei Zhao), full-length *Bos Taurus* αSNAP, full length rat Munc18-1, and a rat Munc13-1 fragment spanning the $C_1C_2$BMUNC$_2$C regions (529–1725 Δ1408–1452). Expression and purification of the corresponding proteins were performed as previously reported (*Chen et al., 2006*; *Chen et al., 2002*; *Dulubova et al., 1999*; *Liang et al., 2013*; *Liu et al., 2017*; *Ma et al., 2011*; *Ma et al., 2013*; *Xu et al., 2013*; *Zhao et al., 2015*) with the modifications described below. His$_6$-full-length syntaxin-1A was induced with 0.4 mM IPTG and expressed overnight at 25°C. Purification was done using Ni-NTA resin (Thermo Fisher) in 20 mM Tris pH 7.4, 500 mM NaCl, 8 mM imidazole, 2% Triton X-100, and 6M urea followed by elution in 20 mM Tris pH 7.4, 500 mM NaCl, 400 mM imidazole, and 0.1% DPC. The His$_6$ tag was removed using thrombin cleavage, followed by size exclusion chromatography on a Superdex 200 column (GE 10/300) in 20 mM Tris pH 7.4, 125 mM NaCl, 1 mM TCEP, 0.2% DPC (*Liang et al., 2013*). GST-syntaxin-1A 2–253 was induced with 0.4 mM IPTG and expressed overnight at 25°C. Purification was done using glutathione sepharose resin (GE) followed by

thrombin cleavage of the GST-tag and anion exchange chromatography on a HiTrap Q column (GE) in 25 mM Tris pH 7.4, 1 mM TCEP using a linear gradient from 0 mM to 1000 mM NaCl. GST-Syb49-93 was induced with 0.4 mM IPTG and expressed overnight at 23°C. Purification was done using glutathione sepharose resin (GE) followed by cleavage of the GST-tag and size exclusion chromatography on a Superdex 75 column (GE 16/60) in 20 mM Tris pH 7.4, 125 mM NaCl. $His_6$-full-length complexin-1 was induced with 0.5 mM IPTG and expressed for 4 hr at 37°C. Purification was done using Ni-NTA resin followed by TEV cleavage of the $His_6$-tag and size exclusion chromatography on a Superdex 75 column (GE 16/60) in 20 mM Tris pH 7.4, 125 mM NaCl, 1 mM TCEP. $His_6$-full-length NSF was induced with 0.4 mM IPTG and expressed overnight at 20°C. Purification was done in 5 steps (*Zhao et al., 2015*): i) Ni-NTA affinity chromatography; ii) size exclusion chromatography of hexameric NSF on a Superdex S200 column (GE 16/60) in 50 mM Tris pH 8.0, 100 mM NaCl, 1 mM ATP, 1 mM EDTA, 1 mM DTT, and 10% glycerol; iii) TEV cleavage of the $His_6$-tag and monomerization with apyrase during 36 hr dialysis with nucleotide-free buffer; iv) three rounds of size exclusion chromatography to separate monomeric and hexameric NSF (re-injecting the latter) on a Superdex S200 column (GE 16/60) in 50 mM NaPi pH 8.0, 100 mM NaCl, 0.5 mM TCEP; and v) reassembly of the NSF monomers and size exclusion chromatography of reassembled hexameric NSF on a Superdex S200 column (GE 16/60) in 50 mM Tris pH 8.0, 100 mM NaCl, 1 mM ATP, 1 mM EDTA, 1 mM DTT, and 10% glycerol. For experiments requiring the use of a non-hydrolyzable analog of ATP, reassembly of monomeric NSF was done in the presence of ATPγS followed by size exclusion chromatography of the hexamer in a similar buffer as before substituting ATPγS for ATP. $His_6$-Munc13-1 $C_1C_2BMUNC_2C$ (529–1725 Δ1408–1452) was induced with 0.5 mM IPTG and expressed overnight at 16°C. Purification was done using Ni-NTA resin (Thermo Fisher) followed by thrombin cleavage of the $His_6$-tag and anion exchange chromatography on a HiTrap Q column (GE) in 20 mM Tris pH 8.0, 10% glycerol, 1 mM TCEP using a linear gradient from 0 to 500 mM NaCl.

## Mutant proteins

All mutations were performed using QuickChange site-directed mutagenesis (Stratagene). These include the S186C mutation in full length syntaxin-1A (C145A, C271A, C272A) and in syntaxin-1A 2–253 (C145A), the M71D,L78D mutation in full-length SNAP-25A (C84S, C85S, C90S, C92S), the L26C mutation in full-length synaptobrevin (C103A), and the F27S,F28S and K122E,K163E mutations in αSNAP. For synaptobrevin L26C, the construct was cloned into a pet28A vector with an N-terminal $His_6$ tag for soluble expression. All mutant proteins were purified as the wild type proteins.

## Labeling proteins with Alexa Fluor 488 and tetramethylrhodamine

Single cysteine mutants were labeled with Alexa Fluor 488 (Alexa488, for full length synaptobrevin L26C) or with tetramethylrhodamine (TMR, for full-length syntaxin-1A S186C and syntaxin-1A 2–253 S186C) using maleimide reactions (Thermo Fisher). Full length synaptobrevin L26C was first buffered exchanged to 20 mM Tris pH 7.4, 150 mM NaCl, 1 mM TCEP, 1% octyl β-glucopyranoside (β-OG) using a PD Miditrap G25 column to provide buffer conditions that allow the reactive thiol group to be sufficiently nucleophilic so that they exclusively react with the dye. Buffered exchanged proteins at a concentration of 75 μM were incubated with a 20-fold excess of dye for 2 hr at room temperature. Unreacted dye was separated from the labeled protein through cation exchange chromatography on a HiTrap SP column (GE) in 25 mM NaAc pH 5.5, 1 mM TCEP, 1% β-OG using a linear gradient from 0 to 1000 mM NaCl. Full length Syntaxin S186C and Syntaxin 2–253 S186C were tagged with tetramethylrhodamine (Thermo Fisher) using a similar protocol. After labeling full length syntaxin-1A S186C, unreacted dye was separated from the labeled protein though anion exchange chromatography on a HiTrap Q column (GE) in 20 mM Tris pH 7.4, 1 mM TCEP, 0.1% DPC using a linear gradient from 0 to 1000 mM NaCl. After labeling syntaxin-1A 2–253 L26C, unreacted dye was separated from the labeled protein using multiple PD Miditrap G25 columns. The concentration of fluorescently tagged proteins was determined using UV-vis absorbance and a Bradford assay.

## Simultaneous lipid mixing and content mixing assays

Assays that simultaneously monitor lipid and content mixing (*Figure 1B,C*) were performed as described in detail in *Liu et al. (2017)* except for a few modifications. Briefly, V-liposomes with full length synaptobrevin contained 39% POPC, 19% DOPS, 19% POPE, 20% cholesterol, 1.5% NBD-PE,

and 1.5% Marina Blue DHPE. T-liposomes with full-length syntaxin-1A and full-length SNAP25 (WT or M71D,L78D mutant) contained 38% POPC, 18% DOPS, 20% POPE, 20% cholesterol, 2% PIP2, and 2% DAG. Dried lipid mixtures were re-suspended in 25 mM HEPES pH 7.4, 150 KCl, 1 mM TCEP, 10% glycerol, 2% β-OG. Purified SNARE proteins and fluorescently labeled content mixing molecules were added to the lipid mixtures to make the syntaxin-1:SNAP25:lipid ratio 1:5:800 and Phycoerythrin-Biotin (4 µM) for T-liposomes, and the synaptobrevin:lipid ratio 1:500 and Cy5-Streptavidin (8 µM) for V-liposomes. The mixtures were incubated at room temperature and dialyzed against the reaction buffer (25 mM HEPES pH 7.4, 150 mM KCl, 1 mM TCEP, 10% glycerol) with 2 g/L Amberlite XAD-2 beads (Sigma) 3 times at 4°C. Proteoliposomes were purified by floatation on a three-layer histodenz gradient (35%, 25%, and 0%) and harvested from the topmost interface. To simultaneously measure lipid mixing from de-quenching of Marina Blue lipids and content mixing from the development of FRET between Phycoerythrin-Biotin trapped in T-liposomes and Cy5-streptavidin trapped in V-liposomes, T-liposomes (0.25 mM lipid) were mixed with V-liposomes (0.125 mM lipid) in a total volume of 200 µL. Acceptor T-liposomes were first incubated with 0.8 µM NSF, 2 µM αSNAP, 2.5 mM $MgCl_2$, 2 mM ATP, 0.1 mM EGTA, and 1 µM Munc18-1 at 37°C for 25 min. They were then mixed with donor V-liposomes, 0.5 µM Munc13-1 $C_1C_2BMUNC_2C$, and 1 µM excess SNAP25. All experiments were performed at 30°C and 0.6 mM $Ca^{2+}$ was added at 300 s. The fluorescence signal from Marina Blue (excitation at 370 nm, emission at 465 nm) and Cy5 (excitation at 565 nm, emission at 670 nm) were recorded to monitor lipid and content mixing, respectively. At the end of the reaction, 1% β-OG was added to solubilize the liposomes and the lipid mixing data were normalized to the maximum fluorescence signal. Most experiments were performed in the presence of 5 µM streptavidin, and control experiments without streptavidin were performed to measure the maximum Cy5 fluorescence after detergent addition for normalization of the content mixing data. Analogous procedures were followed for the lipid and content mixing assays of *Figure 3—figure supplement 2*, except that we used Vsyt1-liposomes instead of V-liposomes. The V-Syt1 liposomes contained 41% POPC, 6.8% DOPS, 29.2% POPE, 20% cholesterol, 1.5% NBD-PE, and 1.5% Marina, as well as full-length synaptobrevin and synaptotagmin-1 (57–421) at 1:10,000 and 1:1000 ratios with the lipids, respectively. Fusion reactions were performed with a 1:4 ratio of VSyt1-to T-liposomes as the trans-SNARE complex assembly assays of *Figure 3E*.

## FRET assays to monitor trans-SNARE complex assembly and disassembly

Reconstituted liposomes were made similarly to those used for the lipid and content mixing assay. V-liposomes with full-length synaptobrevin L26C-Alexa488 contained 42% POPC, 19% DOPS, 19% POPE, and 20% cholesterol. VSyt1-liposomes with Synaptotagmin 57–421 and full length synaptobrevin L26C-Alexa488 contained 43% POPC, 6.8% DOPS, 30.2% POPE, and 20% cholesterol. T-liposomes with full-length syntaxin-1A S186C-TMR and full length SNAP25-A M71D, L78D mutant contained 38% POPC, 18% DOPS, 20% POPE, 20% cholesterol, 2% PIP2, and 2% DAG (note that for selected experiments WT SNAP-25 was used instead of the mutant). Dried lipid mixtures were re-suspended in 25 mM HEPES pH 7.4, 150 KCl, 1 mM TCEP, 2% β-OG. Purified SNARE proteins were added to the lipid mixtures to make the syntaxin-1:SNAP25:lipid ratio 1:5:800 for T-liposomes, the synaptobrevin:lipid ratio 1:10,000 for V-liposomes, and the synaptotagmin-1:synaptobrevin:lipid ratio 1:0.1:1000 for VSyt1-liposomes. The mixtures were incubated at room temperature and dialyzed against the reaction buffer (25 mM HEPES pH 7.4, 150 mM KCl, 1 mM TCEP) with 2 g/L Amberlite XAD-2 beads (Sigma) 3 times at 4°C. All FRET experiments were performed at 37°C on a PTI Quantamaster 400 spectrofluorometer (T-format) equipped with a rapid Peltier temperature controlled four-position sample holder. All slits were set to 1.25 mm. A GG495 longpass filter (Edmund optics) was used to filter scattered light. SNARE complex formation was measured by the development of FRET between Alexa488-Synaptobrevin on V- or VSyt1-liposomes and TMR-syntaxin-1 on T-liposomes or TMR-syntaxin-1 (2–253). Typically, four parallel reactions were monitored simultaneously, which allowed performance of multiple experiments under various conditions (each in triplicate) with the same fresh liposome preparations in the same day.

For kinetic traces, the fluorescence signal at 518 nm (excitation at 468 nm) was recorded to monitor the Alexa488 donor fluorescence intensity. The signal of donor V-or VSyt1-liposomes (0.0625 mM lipid) in reaction buffer containing 0.1 mM EGTA, 2.5 mM $MgCl_2$, 2 mM ATP and various additions was first recorded for 180 s to check for signal stability and then either acceptor T-liposomes

(0.25 mM lipid) or soluble acceptor TMR-syntaxin-1 2–253 (300 nM) were added. The initial additions included the following in different combinations as specified in the Figures and their legends: 10 μM Syb49-93, 2 μM complexin-1, 1 μM synaptotagmin-1 $C_2AB$, 0.6 mM $Ca^{2+}$, 1 μM Munc18-1, and 0.3 μM Munc13-1 $C_1C_2BMUNC_2C$. For experiments where disassembly was tested after recording an assembly reaction (*Figures 4* and *6B–D*), 2 μM αSNAP and 0.4 μM NSF were added at the indicated time points. Because adding these reagents to four reactions running in parallel took more than one minute and disassembly reactions are relatively fast at this time scale, we did not attempt to monitor the kinetics of disassembly and focused on the overall amount of disassembly. For homogeneity among the reactions, we let a short amount of time pass before starting to monitor the donor fluorescence intensity again, which thus was re-initiated 2–5 min after the addition of NSF-αSNAP. For experiments designed to test the assembly of cis- or trans-SNARE complexes in the presence of NSF-αSNAP (*Figures 3* and *6A*), 0.1 mM EGTA, 2.5 mM $MgCl_2$, 2 mM ATP, 2 μM αSNAP and 0.4 μM NSF were mixed with the V- or VSyt1-liposomes from the start, together with the corresponding additional proteins. For some experiments, $Ca^{2+}$ (0.6 mM) was added at the time points indicated in the Figures. In the experiments of *Figures 3C,D,F* and 240 μM $Ca^{2+}$ was added at 2 min and 500 μM EGTA at the indicated time points. Data points were collected (1 s acquisition) every 20 s for 30 min for reactions where saturation was reached, and for longer times for slower reactions. Only a small amount of photobleaching of the donor was observed under these conditions in control experiments with donor alone.

Pre-formed trans-SNARE complexes were made by incubating V-liposomes, T-liposomes, SNAP25m, Synaptobrevin 49–93, and 0.1 mM EGTA together for 5 hr at 37°C. For experiments with WT SNAP-25 or samples with VSyt1 liposomes, these reagents were mixed together for 24 hr at 4°C. Various combinations of the proteins listed above, as well as 2.5 mM $MgCl_2$ and 2 mM ATP, were added to the pre-formed trans-SNARE complex and incubated for 5 min at 37°C. An emission scan was then collected (excitation 468 nm, emission from 490 nm to 700 nm) to detect how much trans-SNARE complex was formed. Two μM αSNAP and 0.4 μM NSF were then added to each reaction and allowed to incubate for 5 min at 37°C to disassemble the trans-SNARE complex. A second wavelength scan was then collected to determine how much of the complex was disassembled. All experiments were repeated at least 3 times with a single preparation and the results were verified in multiple experiments with different preparations. For some control experiments, ATPγS (2 mM) was used instead of ATP, or $Mg^{2+}$ was replaced with EDTA (1 mM).

## Cryo-electron microscopy

Samples of pre-formed trans-SNARE complexes between V- and T-liposomes were prepared by incubating them with Syb49-93 for five hours at 37°C as for the assays used to measure protection against disassembly by NSF-αSNAP. Cryo-EM grids were prepared by applying 3 μL of the sample solution to a negatively glow discharged Lacey carbon copper grid (200-mesh; Electron Microscopy Sciences) and blotted for 4.0 s under 100% humidity at 4°C before plunge-freezing in liquid ethane using a Mark IV Vitrobot (FEI). Micrographs were acquired on a Talos Arctica microsope (FEI) operated at 200 kV with a K2 Summit direct electron detector (Gatan). A nominal magnification of 11,000 was used for imaging, and 20 dose-fractionation frames were recorded over a 10 s exposure at a dose rate of 2.1 electrons/ $Å^2$/s for each micrograph. Motion correction was performed using the MotionCorr2 program (*Zheng et al., 2017*).

## Acknowledgments

We thank Minglei Zhao for providing the plasmid to express NSF, and Bradley Quade for providing purified proteins. We also thank the reviewers of this paper for their constructive criticisms, which have helped to considerably improve its quality. Cryo-EM data were collected at the University of Texas Southwestern Medical Center (UTSW) Cryo-Electron Microscopy (Cryo-EM) Facility that is funded in part by the CPRIT Core Facility Support Award RP170644. We thank Daniel Stoddard for training and maintenance of the UTSW Cryo-EM Facility. Eric Prinslow was supported by NIH Training Grant T32 GM008297. This work was supported by grant I-1304 from the Welch Foundation (to JR) and by NIH Research Project Award R35 NS097333 (to JR).

## Additional information

### Funding

| Funder | Grant reference number | Author |
|---|---|---|
| National Institute of Neurological Disorders and Stroke | R35 NS097333 | Josep Rizo |
| Welch Foundation | I-1304 | Josep Rizo |
| National Institute of General Medical Sciences | T32 GM008297 | Eric A Prinslow |

The funders had no role in study design, data collection and interpretation, or the decision to submit the work for publication.

### Author contributions

Eric A Prinslow, Conceptualization, Formal analysis, Investigation, Methodology, Writing—review and editing; Karolina P Stepien, Formal analysis, Investigation, Writing—review and editing; Yun-Zu Pan, Investigation, Methodology, Writing—review and editing; Junjie Xu, Formal analysis, Investigation, Methodology; Josep Rizo, Conceptualization, Formal analysis, Supervision, Funding acquisition, Investigation, Methodology, Writing—original draft, Project administration

### Author ORCIDs

Josep Rizo (iD) http://orcid.org/0000-0003-1773-8311

### Decision letter and Author response

Decision letter https://doi.org/10.7554/eLife.38880.024
Author response https://doi.org/10.7554/eLife.38880.025

## Additional files

### Supplementary files

• Transparent reporting form
DOI: https://doi.org/10.7554/eLife.38880.022

### Data availability

Representative examples of all the data generated and analyzed during this study are included in the manuscript and supporting files.

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
