## [Decision Letter]

Thank you for submitting your article "Multiple factors synergistically protect trans-SNARE complexes against disassembly by NSF and aSNAP" for consideration by *eLife*. Your article has been reviewed by three peer reviewers, and the evaluation has been overseen by Axel Brunger as the Reviewing Editor and Randy Schekman as the Senior Editor. The following individual involved in review of your submission has agreed to reveal their identity: Yongli Zhang (Reviewer #2).

Summary:

This study addresses the effects of NSF, αSNAP, synaptotagmin, CA, Munc 18, Munc 13, and complexin on the assembly, disassembly, and stability of trans-SNARE complexes.

Overall, this study contains a mixture of results that were already known (such as the ability of NSF to disassemble trans SNARE complexes, Yavuz et al.., 2018, or the effects of various factors on trans SNARE complex assembly, Ma et al., 2013), along with new results, in particular those shown in Figures 3 and 6. This mixture of some old and some new results makes the paper somewhat difficult to read and reduces its impact.

In addition, the reviewers and reviewing editor identified a number of serious concerns as outlined below. In the current form, the manuscript is not therefore not acceptable. However, we are open to consider a re-written manuscript that focusses on new and interesting findings and address the concerns raised below.

Major comments:

1) One major conclusion is that NSF/SNAP can disassemble trans-SNARE complexes. As the authors acknowledge, however, the novelty of this conclusion is unfortunately undermined by a recent report from the Jahn lab (Yavuz, 2018). Another major conclusion is that Munc18 and Munc13 together support trans-SNARE complex assembly in the presence of NSF/SNAP. This is important, and consistent with the results presented in this manuscript, but it was already well established by previous work, especially from the Rizo lab (especially Ma et al., 2013). Please re-write the paper, focusing on new insights, and remove material that is similar to previously published work (or relegate it to a supplement).

2) Throughout the work, the authors used mutant SNAP-25 with two mutations in its C-terminal region, which attenuates SNARE zippering and stabilizes the partially zippered SNARE conformation for trans-SNAREs. The authors should address the impact of such mutations on the major conclusions derived for wild-type SNAREs. A clarification on this issue is important, which affects comparison of this work with previous work (especially, Weber et al., 2000).

3) t-SNARE (3Q) proteoliposomes tend to form a dead-end conformation of 2 syntaxins (2Qa) associated with 1 SNAP-25 (QbQc); to prevent this, the authors start with a SNARE domain fragment Syb49-93 which can enter the complex to form [Sb49-93]:Syntaxin:SNAP25. The synaptobrevin coming in trans purportedly displaces the Syb49-93 and gives the authentic parallel trans-SNARE complex. For clarity, the authors really must expand their presentation of this in Figure 2A. The cartoon in Figure 2A does not present where and how Syb49-93 binds in the initial v-SNARE (3Q) complex, and they add a second synthetic SNARE fragment as well, Syb29-93, to prevent trans-SNARE complex reassembly. If as claimed (Figure 2B) NSF and SNAP are disassembling trans-SNARE complex, why would the reassembly of that complex be a problem? Wouldn't it just be again disassembled by NSF/SNAP? Presumably they found that Syb29-93 was need to keep the bulk of the system in a NSF/SNAP-induced disassembled structure at steady-state. In sum, the proteoliposomes the authors are using have 2 SNARE-domain mutations, introducing charged residues, to block the completion of zippering, multiple mutations to eliminate the cysteines, two mutations to introduce not just Cys residues at the desired positions but then stoichiometric derivatization of these Cys residues with fluorophores, and there are two (2) extra SNARE domain peptides introduced as well. Moreover, starting the system with syntaxin in complex with Munc18-1 and then using Munc13, the fundamental insight of their Science paper of 2013 (Ma et al.), should get around the need for Syb49-93 and Syb29-93.

4) Figure 2B: The fluorescence emission spectrum of the v-SNARE liposomes is clear, just one peak at about 525nm (black), and there is a big reduction in this peak due to quenching when the t-SNARE proteoliposomes are added (red) and a new FRET peak appears at around 580 nm. Addition of NSF + αSNAP relieves most of the quenching at 525nm without removing most of the presumptive FRET peak at 580. Presumably, this is due to direct excitation of the acceptor dye, as pointed out by the authors in a parenthetical note in the fourth paragraph of the subsection “Munc18-1, Munc13-1, complexin-1 and the synaptotagmin-1 C2 domains stabilize trans-SNARE complexes”. Please measure this effect for by collecting an emission spectrum of the same sample preparation with only acceptor dyes, but no donor dyes, using the same excitation wavelength that was used in Figure 2. Of course, all the data need to be viewed in the context of this direct donor excitation, and ideally, all the data need to be corrected for it, although this is difficult with bulk fluorescence experiments. Another complication in interpretation of such bulk fluorescence experiments is that differences could be due to affecting the acceptor dye by molecular interactions rather than trans SNARE complex disassembly per se. For example, might the green curve in Figure 2B be due to an altered distance and conformation change in the 2 fluorophores when α-SNAP and NSF bind rather than to true disassembly? Thus, controls are essential (e.g., Mg^2+^ vs EDTA, and no ATP, ATP, or ATPγS). Comparison of ATP vs. ATPγS should be done throughout.

5) Related to the above concern, in Figure 3 the major effect of synaptotagmin, Munc18, Mun13, and complexin seems to be in suppressing the donor:acceptor ratio before the NSF/SNAP is added. In this figure, trans-SNARE complex spontaneous assembly was first allowed, then synaptotagmin, Munc18, Mun13, and/or complexin added, then NSF/SNAP. It is possible that these might substantially modify the environment of the fluorophores and affect FRET as much as they actually affect trans-SNARE disassembly. As mentioned above, controls need to be carried out to roll out such effects on the fluorophores.

6) The paper would benefit tremendously if the authors could complement their fluorescence study with another assay of SNARE association, such as pull-down experiments. Also, it would strengthen the study to know which combinations of factors would give fusion (they use an assay of protected fluorescent protein mixing in Figure 1) if they used wild-type or less permanently arrested SNAREs. In Figure 2E, there is a striking change in donor fluorescence, which I presume represents trans-SNARE complex assembly, which depends on Munc18, Munc13, and calcium. With unblocked SNAREs, does this lead to fusion?

7) A key question is how NSF/α-SNAP distinguish trans- and cis-SNARE complexes. The authors argue that α-SNAP cannot bind to two membranes well as in trans-SNARE complex, which prevents NSF/α-SNAP from disassembling trans-SNARE complexes. However, membrane binding by α-SNAP is not required for SNARE disassembly, as NSF/α-SNAP efficiently disassemble cytoplasmic SNARE complexes. Thus, it is possible that NSF/alpha-SNAP recognize certain SNARE conformational differences between trans- and cis-SNAREs. For example, as is previously proposed (Minglei Zhao et al., Nature, 2015), trans-SNAREs may be partially zippered. The authors are encouraged to reveal more insights into the key question by more experiments if possible.

8) The authors claim, in the title as well as the text, that Munc18, Munc13, synaptotagmin, and complexin act synergistically to protect trans-SNARE complexes from NSF/SNAP. However, this is not obvious based on the data presented in Figure 3C/D, where greater-than-additive effects are not observed at least for most combinations. Complexin and synaptotagmin on their own, at the concentration used (but see the next point #9), have only modest activity, which might simply reflect the ability of any binding partner to get in the way of NSF/SNAP. In the absence of calcium, there is a bit of additional protection afforded by complexin (but not by synaptotagmin) when Munc18 and Munc13 are already present.

9) The claim that complexin does not prevent disassembly of cis SNARE complexes (Figure 4) is at odds with other results in the literature. Previously, at a SNAP:complexin ratio of ~ 18:1, complexin had no effect (Pabst et al., 2000), whereas at a ratio of ~ 3:1, complexin had a substantial effect on NSF-mediated disassembly of the soluble SNARE complex (Winter et al., 2009). More recently, concentration-dependent complexin inhibition of NSF-mediated disassembly of the cis SNARE complex was observed (Choi et al., 2018). Please perform the experiments shown in Figures 3, 4, 6 at different complexin concentrations, i.e., different complexin:SNAP molar ratios.

10) Figures 3 and 6 show protection of disassembly of trans SNARE complexes by a variety of factors. A key difference between experiments in these two figures relates to synaptotagmin. The C2AB fragment is used in Figure 6, whereas full-length synaptotagmin is used in Figure 6. Why? Please redo all experiments in Figure 3 with membrane anchored synaptotagmin (as in Figure 6).

[Editors' note: further revisions were requested prior to acceptance, as described below.]

Thank you for resubmitting your work entitled "Multiple factors protect neuronal trans-SNARE complexes against disassembly by NSF and aSNAP" for further consideration at *eLife*. Your revised article has been favorably evaluated by Randy Schekman as the Senior Editor, Axel Brunger as the Reviewing Editor, and three reviewers.

The manuscript has been improved but there are some remaining issues that need to be addressed before acceptance, as outlined below.

Major comments:

1) The data make a convincing case that complexin can bind trans-SNARE complexes to block NSF/SNAP-driven disassembly. This conclusion, moreover, seems entirely plausible based on previous findings, including high-resolution structures. The role(s) of Munc18/13 on inhibiting trans SNARE complex disassembly is more problematic. It was known that, working together, they powerfully promote trans-SNARE complex assembly. The simplest model, call it Model I, is that Munc18/13 assembly activity simply opposes NSF/SNAP disassembly activity. Model I does not require that Munc18/13 interact with trans-SNARE complexes. By contrast the author's model, call it Model II, is that Munc18/13 remains associated with assembled trans-SNARE complexes. In so doing, Munc18/13 blocks NSF/SNAP from binding, and therefore disassembling, trans-SNARE complexes.

Moreover, NSF/SNAP plays a dual role: not only does it catalyze SNARE disassembly, but it also promotes Munc18/13-mediated SNARE assembly by rescuing SNAREs that are kinetically trapped in off-pathway, dead end products. No one, of course, knows this better than the authors, who indeed point out this effect in Figure 4B. Nevertheless, the dual role of NSF/SNAP further complicates the task of unambiguously distinguishing models I and II by using the author's assays.

The revised manuscript does not conclusively rule out Model I since every manipulation that strengthens the postulated 'protective' activity of Munc18/13, including the addition of Ca^2+^, also increases its assembly activity. Even the authors, upon occasion, seem to adopt the language of Model I. For example, "[T]he contrast of the results obtained with cis-SNARE complexes with those observed with trans-SNARE complexes provides a dramatic demonstration of how the apposition of two membranes tilts the balance in favor of SNARE complex assembly, whereas disassembly dominates on a single membrane".

What would seem to be needed for distinguishing Model I from Model II is some manipulation or mutation that separates the proposed Munc18/13's protective function from its assembly function. For example, this could be accomplished by determining if Munc18 and Munc13 are both associated with trans SNARE complexes prior to NSF/aSNAP-mediated disassembly. However, we realize that such a direct observation may not be possible with the author's assays. At the minimum, we request a more balanced discussion of these points, presenting both models I and II as being consistent with the data presented in this paper and other available data, as well as adjusting the title, Abstract and other relevant parts of the text.

2) An additional model to explain the data is as follows (call it Model III): Munc18/13 catalyze the proper assembly of the SNARE complex, and thereby, supercomplexes with complexin and synaptotagmin. Once such properly assembled supercomplexes are situated between membranes, it might be difficult for SNAP/NSF to bind to the trans-SNARE complex even if Munc13/Munc18 are not associated with such supercomplexes anymore. The absence of Munc18/Munc13 may lead to a mixture of properly and improperly assembled trans-SNARE complexes where the latter are more readily disassembled by NSF/SNAP. Please discuss this possible model as well.

3) The authors performed additional experiments to test inhibition of cis-SNARE complexes by complexin (Figure 6, Figure 6—figure supplement 1, Results section "Disassembly of cis-SNARE complexes”; Discussion section, seventh paragraph), and state: "It is plausible that the differences observed with the results of Winter et al.. 2009, and Choi et al., 2018, arose because syntaxin-1 did not include the N-terminal region containing the Habc domain in both of these studies, and in the latter NSF- αSNAP might have been less active because of the absence of membranes." However, contrary to what is stated here, the Winter et al. experiments did include a membrane. The reviewers are not convinced that there is any discrepancy between the data presented in the present work and the previous work by Winter et al., and Choi et al. since the experiment shown in Figure 6—figure supplement 1 does not have the necessary time resolution to discern the effect of complexin on the kinetics of NSF/aSNAP-mediated SNARE complex disassembly observed by Winter et al. and Choi et al.

Moreover, the drawing in Figure 6—figure supplement 1 is somewhat misleading since upon injection of NSF/aSNAP, at least a minute (or more) passes before the fluorescence intensity measurements continue. Thus, this break period should be clearly marked in the figure and explained in the figure caption or the Materials and methods. The presentation and discussion of the results in the subsection “Disassembly of cis-SNARE complexes” and the Discussion, should be adjusted to present a unified view of the action of complexin on slowing the kinetics of NSF/aSNAP-mediated for both cis and trans SNARE complexes (i.e., complexin slows the kinetics in both cases to different degrees).

---

## [Author Response]

Summary:This study addresses the effects of NSF, αSNAP, synaptotagmin, CA, Munc 18, Munc 13, and complexin on the assembly, disassembly, and stability of trans-SNARE complexes.Overall, this study contains a mixture of results that were already known (such as the ability of NSF to disassemble trans SNARE complexes, Yavuz et al., 2018, or the effects of various factors on trans SNARE complex assembly, Ma et al., 2013), along with new results, in particular those shown in Figures 3 and 6. This mixture of some old and some new results makes the paper somewhat difficult to read and reduces its impact.In addition, the reviewers and reviewing editor identified a number of serious concerns as outlined below. In the current form, the manuscript is not therefore not acceptable. However, we are open to consider a re-written manuscript that focusses on new and interesting findings and address the concerns raised below.

We fully understand these concerns and appreciate the opportunity to address them in the revised manuscript. There are a few points that we would like to emphasize about the complexity of the system and the difficulty of investigating trans-SNARE complexes between two membranes before we provide answers to the specific concerns.

SNAREs are highly abundant in the vesicle and plasma membranes, but assembly of many trans-SNARE complexes may lead to uncontrolled membrane fusion and perhaps to fusion pathways that are not physiologically relevant. Cryo-EM images of liposome fusion reactions with SNAREs have shown extended membrane interfaces [e.g. Hernandez et al. Science *336*, 1581 (2012)] that we believe are unlikely to occur in the pathway to synaptic vesicle fusion. Indeed, inclusion of other proteins affects the morphology of the membrane interfaces [e.g. Bharat et al. EMBO Rep. *15*, 308 (2014)] and favors ‘point contact’ interfaces that are more likely to be physiologically relevant [Gipson et al., 2017]. We believe that Munc13-1 and other components of the release machinery limit the number of trans-SNARE complexes that are formed before priming, but it is difficult to faithfully recapitulate the primed state of synaptic vesicles with a limit set of proteins.

The interpretation of the data is further complicated by the finding that some trans-SNARE complexes can be disassembled by NSF-αSNAP while others are NSF-αSNAP resistant, as described in the recent Yavuz et al. 2018., which attributes such resistance to formation of the extended liposome interfaces mentioned above. In our experiments we used a very low synaptobrevin-to-lipid ratio (1:10,000) in part to maximize the percent of synaptobrevin molecules that engage in trans-SNARE complex formation, thus optimizing the decrease in donor fluorescence, and in part to avoid accumulation of too many trans-SNARE complexes between each membrane-membrane interface. Even with this design, we still observe that 50% of trans-SNARE complexes are resistant to NSF-αSNAP, and cryo-EM images of liposomes with pre-formed trans-SNARE complexes still show some extended interfaces between liposomes (new Figure 2—figure supplement 3), albeit smaller than those observed by Yavuz et al., 2018. In the paper we suggest that the trans-SNARE complexes that can be disassembled by NSF-αSNAP may be more representative of those present in primed synaptic vesicles (Discussion, second paragraph), which is a speculation but correlates with the finding that synaptic vesicles can be de-primed and that de-priming is NEM-sensitive [He et al., 2017].

The observation of NSF-αSNAP-resistant trans-SNARE complexes provides a plausible explanation for the results of Weber et al., 2000, which showed that lipid mixing between V- and T-liposomes could occur in the presence of NSF-αSNAP only if the liposomes were pre-incubated (we now suggest this explanation in the second paragraph of the Discussion). We have tried to use liposome fusion assays to investigate this issue but so far have not obtained sufficiently conclusive results. We believe that resolving this issue is beyond the scope of this paper. In the revised manuscript we do include new fusion assays (Figure 3—figure supplement 2) that illustrate that trans-SNARE complex assembly does not necessarily lead to membrane fusion (see points 1 and 6 below).

We would also like to point out that we attempted to perform pulldown assays, as suggested by the reviewers, and also co-sedimentation assays between heavy V-liposomes containing sucrose and T-liposomes, to obtain data that could complement the FRET results, but it soon became clear that both approaches cannot yield meaningful results because Munc13-1 and synaptotagmin-1 can bridge membranes even if there are no trans-SNARE complexes. We hope that the reviewers will agree that the FRET approach that we used is the most appropriate for direct measurement of transSNARE complex assembly and disassembly, and that, with the controls that were requested and that extensive additional data that we acquired to address the reviewer concerns, this is a strong story that provides key insights into how central components of the release machinery protect trans-SNARE complexes against disassembly by NSFαSNAP.

Major comments:1) One major conclusion is that NSF/SNAP can disassemble trans-SNARE complexes. As the authors acknowledge, however, the novelty of this conclusion is unfortunately undermined by a recent report from the Jahn lab (Yavuz, 2018). Another major conclusion is that Munc18 and Munc13 together support trans-SNARE complex assembly in the presence of NSF/SNAP. This is important, and consistent with the results presented in this manuscript, but it was already well established by previous work, especially from the Rizo lab (especially Ma et al., 2013). Please re-write the paper, focusing on new insights, and remove material that is similar to previously published work (or relegate it to a supplement).

In the revised manuscript we do not present the disassembly of trans-SNARE complexes by NSF-αSNAP as a conclusion of the paper and we have increased the emphasis on the newest aspect of our results, namely the analysis of protection of trans-SNARE complexes against disassembly by NSF-αSNAP. However, we would like to clarify that, while it is generally assumed that trans-SNARE complex assembly is required for lipid and content mixing, and in the Ma et al., 2013 paper we did propose that Munc18-1 and Munc13-1 orchestrate SNARE complex assembly in an NSF-αSNAP resistant manner, we did not really measure trans-SNARE complexes in that paper. The contrast between trans-SNARE complex assembly and fusion is now illustrated by comparing the transSNARE complex assembly assays between VSyt1- and T-liposomes of Figure 3E (red curve), where Ca^2+^-independent assembly is observed, and the fusion assays shown in the new Figure 3—figure supplement 2, where there is no fusion before addition of Ca^2+^. Moreover, the analysis of protection against disassembly is tightly linked to the analysis of trans-SNARE complex assembly in the presence of NSF and αSNAP because the former may depend on the latter. Hence, we cannot really separate the assembly data from the disassembly results.

2) Throughout the work, the authors used mutant SNAP-25 with two mutations in its C-terminal region, which attenuates SNARE zippering and stabilizes the partially zippered SNARE conformation for trans-SNAREs. The authors should address the impact of such mutations on the major conclusions derived for wild-type SNAREs. A clarification on this issue is important, which affects comparison of this work with previous work (especially, Weber et al., 2000).

We have performed some experiments with WT SNAP-25 instead of SNAP-25m (Figure 2—figure supplement 4B), verifying that the SNAP-25 mutation does not impair trans-SNARE complex disassembly by NSF-αSNAP. Moreover, we now also show that the mutation in SNAP-25m does not interfere with disassembly of cis-SNARE complexes (new Figure 6—figure supplement 1B, C).

3) t-SNARE (3Q) proteoliposomes tend to form a dead-end conformation of 2 syntaxins (2Qa) associated with 1 SNAP-25 (QbQc); to prevent this, the authors start with a SNARE domain fragment Syb49-93 which can enter the complex to form [Sb49-93]:Syntaxin:SNAP25. The synaptobrevin coming in trans purportedly displaces the Syb49-93 and gives the authentic parallel trans-SNARE complex. For clarity, the authors really must expand their presentation of this in Figure 2A. The cartoon in Figure 2A does not present where and how Syb49-93 binds in the initial v-SNARE (3Q) complex, and they add a second synthetic SNARE fragment as well, Syb29-93, to prevent trans-SNARE complex reassembly. If as claimed (Figure 2B) NSF and SNAP are disassembling trans-SNARE complex, why would the reassembly of that complex be a problem? Wouldn't it just be again disassembled by NSF/SNAP? Presumably they found that Syb29-93 was need to keep the bulk of the system in a NSF/SNAP-induced disassembled structure at steady-state. In sum, the proteoliposomes the authors are using have 2 SNARE-domain mutations, introducing charged residues, to block the completion of zippering, multiple mutations to eliminate the cysteines, two mutations to introduce not just Cys residues at the desired positions but then stoichiometric derivatization of these Cys residues with fluorophores, and there are two (2) extra SNARE domain peptides introduced as well. Moreover, starting the system with syntaxin in complex with Munc18-1 and then using Munc13, the fundamental insight of their Science paper of 2013 (Ma et al.), should get around the need for Syb49-93 and Syb29-93.

We have improved Figure 2A to help visualizing the effects of Syb49-93. In our original experiments we used Syb29-93 to prevent re-assembly of SNARE complexes, similar to the approach used in Winter et al., 2009. However, we have performed additional experiments and found that Syb29-93 does not affect the results when we add NSFαSNAP to disassemble trans-SNARE complexes, most likely because re-assembly is much slower than disassembly. We have repeated all the protection assays without Syb29-93 and obtained analogous results to those that we described originally. All the data presented in the revised manuscript were obtained without Syb29-93.

We understand the concern about so many other manipulations, but many of them are common in the field and are based on solid grounds. Mutations of native cysteines have been widely used in studies where cysteines were introduced in non-native positions of SNARE proteins to attach fluorescent probes. The positions where we attached probes on synaptobrevin and syntaxin-1 were chosen carefully to minimize the possibility that they might hinder binding of the various factors used in our experiments, based on the available structural information. The protection against disassembly provided by these factors suggests that the probes indeed did not prevent binding.

We did perform experiments where NSF-αSNAP were added from the beginning and trans-SNARE complex assembly was monitored in the presence of different factors, which confirmed that Munc13-1 and Munc13-1 C_1_C_2_BMUNC_2_C are essential for transSNARE complex assembly in the presence of NSF-αSNAP (Figures 3A, B, E). This is the ideal set up to guide the system through the Munc18-1-Munc13-1-dependent pathway of trans-SNARE complex assembly, and these experiments do suggest that Munc18-1 and Munc13-1 C_1_C_2_BMUNC_2_C must somehow protect against disassembly, particularly in the presence of Ca^2+^. However, it was less clear whether Munc18-1 and Munc13-1 C_1_C_2_BMUNC_2_C protect against disassembly in the absence of Ca^2+^, as Ca^2+^-independent assembly was inefficient. We have performed additional experiments where EGTA was added after efficient Ca^2+^-dependent assembly of trans-SNARE complexes to test whether they could be disassembled by the NSF-αSNAP present in the reaction, and found that there is some but limited disassembly (Figures 3C, D). These results support the notion that Munc18-1 and Munc13-1 C_1_C_2_BMUNC_2_C protect against disassembly in the absence of Ca^2+^ and that such protection is enhanced by Ca^2+^. Using this approach, we further show that complexin-1 strongly protects against disassembly (Figures 3D, F).

While these experiments were very informative, we also wanted to test whether Munc18-1, Munc13-1 C_1_C_2_BMUNC_2_C, synaptotagmin-1 and complexin-1 individually or in different combinations can protect trans-SNARE complexes against disassembly. For this purpose, we use the approach of pre-forming trans-SNARE complexes in the absence of any of these proteins and then testing whether incubation of the pre-formed complexes with the proteins prevents disassembly by NSF-αSNAP. Including the Syb4993 peptide in these experiments was critical to accelerate trans-SNARE complex assembly and reach optimal assembly in a reasonable amount of time, as we show now in Figure 2—figure supplement 2. Yavuz et al., 2018 showed that this peptide is released upon trans-SNARE complex assembly; hence, Syb49-93 should not interfere with the disassembly reaction.

4) Figure 2B: The fluorescence emission spectrum of the v-SNARE liposomes is clear, just one peak at about 525nm (black), and there is a big reduction in this peak due to quenching when the t-SNARE proteoliposomes are added (red) and a new FRET peak appears at around 580 nm. Addition of NSF + αSNAP relieves most of the quenching at 525nm without removing most of the presumptive FRET peak at 580. Presumably, this is due to direct excitation of the acceptor dye, as pointed out by the authors in a parenthetical note in the fourth paragraph of the subsection “Munc18-1, Munc13-1, complexin-1 and the synaptotagmin-1 C2 domains stabilize trans-SNARE complexes”. Please measure this effect for by collecting an emission spectrum of the same sample preparation with only acceptor dyes, but no donor dyes, using the same excitation wavelength that was used in Figure 2. Of course, all the data need to be viewed in the context of this direct donor excitation, and ideally, all the data need to be corrected for it, although this is difficult with bulk fluorescence experiments.

We acquired fluorescence emission spectra for separate samples of donor and acceptor (Figure 2—figure supplement 4A). There is indeed some excitation of the acceptor even using the donor excitation wavelength, and the emission intensity is considerable because the acceptor is used in large excess over the donor. We now use the addition of the two separate spectra as the control for ‘no assembly’ (Figure 2B, Figure 2—figure supplements 4B, C). As we explain in point 5, our assessment of the ability of different factors to protect against disassembly now relies on measurements of donor fluorescence at 518 nm, where its emission is highest. The fluorescence emission spectra that we acquired for the acceptor alone showed that the emission of the acceptor at 518 nm is negligible, although there is a small signal intensity at this wavelength that arises from scattering (Figure 2—figure supplement 4A). We did not attempt to correct for this contribution to the intensity at 518 nm because we did not calculate FRET efficiencies and the small contribution from scattering is analogous under the various conditions where we measured donor fluorescence intensities.

Another complication in interpretation of such bulk fluorescence experiments is that differences could be due to affecting the acceptor dye by molecular interactions rather than trans SNARE complex disassembly per se. For example, might the green curve in Figure 2B be due to an altered distance and conformation change in the 2 fluorophores when α-SNAP and NSF bind rather than to true disassembly? Thus, controls are essential (e.g., Mg^2+^ vs EDTA, and no ATP, ATP, or ATPγS). Comparison of ATP vs. ATPγS should be done throughout.

We address the issue of altered distance of conformation in point 5. In the revised manuscript we now present controls for trans-SNARE complex disassembly where Mg^2+^ was replaced by EDTA or ATP was replaced by ATPγS (Figures 2C, 5B, C, Figure 5—figure supplements 1, 4).

5) Related to the above concern, in Figure 3 the major effect of synaptotagmin, Munc18, Mun13, and complexin seems to be in suppressing the donor:acceptor ratio before the NSF/SNAP is added. In this figure, trans-SNARE complex spontaneous assembly was first allowed, then synaptotagmin, Munc18, Mun13, and/or complexin added, then NSF/SNAP. It is possible that these might substantially modify the environment of the fluorophores and affect FRET as much as they actually affect trans-SNARE disassembly. As mentioned above, controls need to be carried out to roll out such effects on the fluorophores.

This is a very important point. First we would like to clarify that Figure 5A of the revised manuscript (which corresponds to Figure 3B in the original paper) shows two spectra, one acquired after incubating pre-formed trans-SNARE complexes with Munc18-1, Munc13-1 C_1_C_2_BMUNC_2_C, synaptotagmin-1 C_2_AB, complexin-1 and Ca^2+^ and another acquired on the same sample after adding NSF-αSNAP. Analogous spectra were acquired with different combinations of these proteins to dissect their abilities to protect against trans-SNARE complex disassembly. Representative spectra for all the different conditions are now shown in Figure 5—figure supplement 1. As we now mention in the manuscript (subsection “Multiple factors stabilize trans-SNARE complexes against disassembly by NSF-αSNAP”, sixth paragraph), all the spectra acquired after incubating trans-SNARE complexes with different combinations of Munc18-1, Munc13-1 C_1_C_2_BMUNC_2_C, synaptotagmin-1 C_2_AB, complexin-1 and Ca^2+^ were very similar, showing that these proteins do not substantially affect the fluorophores in trans-SNARE complexes. However, there were clear differences among the spectra obtained after NSF-αSNAP addition, some of which could arise from proteins affecting the fluorophores rather than disassembly per se. To assess the potential effects of the various proteins on the fluorescence emission spectra of the donor and acceptor in samples as similar as possible to those used to examine protection against disassembly, we prepared control samples with V-liposomes containing Alexa488-labeled synaptobrevin (V*) and T-liposomes containing no fluorescence acceptor (T), as well as controls samples where the V-liposomes did not have a fluorescence donor (V) and T-liposomes with TMR-labeled syntaxin-1 (T*). The spectra obtained with different additions showed that the different proteins did not affect the donor fluorescence substantially (Figure 5—figure supplement 2), but NSF-αSNAP did affect the acceptor fluorescence considerably and this effect was altered under some conditions, particularly by the inclusion of Munc18-1 (Figure 5—figure supplement 3).

As it would be difficult to account for these different effects on the acceptor fluorescence, to obtain a quantitative measure of protection against disassembly we now rely only on donor fluorescence intensities, and calculated the ratio of these intensities after and before NSF-αSNAP addition (Figures 5B, C). Note that in the original manuscript we calculated the ratio between donor and acceptor intensities, which in retrospective was clearly wrong. We are very thankful to the reviewers for helping us to correct this mistake. The overall results that we obtained in terms of the protection afforded by different proteins are similar to those described in the original manuscript, but there is an important distinction in that Munc18-1 alone appeared previously to provide robust protection against disassembly but with the new data such protection appears to be rather limited or none. Nevertheless, Munc18-1 still cooperates with Munc13-1 in protecting against disassembly, particularly in the presence of Ca^2+^, and optimal protection is observed when all four proteins are included, as concluded in the original manuscript.

6) The paper would benefit tremendously if the authors could complement their fluorescence study with another assay of SNARE association, such as pull-down experiments. Also, it would strengthen the study to know which combinations of factors would give fusion (they use an assay of protected fluorescent protein mixing in Figure 1) if they used wild-type or less permanently arrested SNAREs. In Figure 2E, there is a striking change in donor fluorescence, which I presume represents trans-SNARE complex assembly, which depends on Munc18, Munc13, and calcium. With unblocked SNAREs, does this lead to fusion?

We agree that it would be desirable to use other methods to confirm the results from the FRET assays. However, as we explain in our response to the Summary, pulldown assays did not provide useful information because Munc13-1 and synaptotagmin-1 can bridge membranes regardless of the presence of trans-SNARE complexes.

With regard to the change in fluorescence in Figure 2E (now Figure 4B), this Ca^2+^dependent change does correlate with the Ca^2+^-dependence of liposome fusion in our reconstitution assays (Ma et al., 2013; Liu et al., 2016), although more closely related experiments are those where NSF-αSNAP were included from the beginning (red traces in what are now Figure 3A, for experiments with V-liposomes and Figure 3E for experiments with VSyt1-liposomes). To examine whether there is fusion under conditions that are more similar to those used for the trans-SNARE complex assembly assays, we performed fusion assays using VSyt1- and T-liposomes with the same protein-to-lipid ratios as in the assembly assays, but using WT SNAP-25 instead of the SNAP-25 mutant to allow fusion to occur. The results show that there is no fusion before Ca^2+^ addition and efficient but slow fusion upon Ca^2+^ addition (Figure 3—figure supplement 2); in contrast, trans-SNARE complex assembly does occur in the absence of Ca^2+^ and is very fast upon addition of Ca^2+^ (Figure 3E). These results show that transSNARE complex assembly does not necessarily lead to membrane fusion and that the FRET assay that we have developed provides a much better tool to analyze protection of trans-SNARE complexes against disassembly by NSF-αSNAP than liposome fusion assays.

7) A key question is how NSF/α-SNAP distinguish trans- and cis-SNARE complexes. The authors argue that α-SNAP cannot bind to two membranes well as in trans-SNARE complex, which prevents NSF/α-SNAP from disassembling trans-SNARE complexes. However, membrane binding by α-SNAP is not required for SNARE disassembly, as NSF/α-SNAP efficiently disassemble cytoplasmic SNARE complexes. Thus, it is possible that NSF/alpha-SNAP recognize certain SNARE conformational differences between trans- and cis-SNAREs. For example, as is previously proposed (Minglei Zhao et al., Nature, 2015), trans-SNAREs may be partially zippered. The authors are encouraged to reveal more insights into the key question by more experiments if possible.

This is an important issue but we believe that obtaining definitive data to address this question is outside the scope of this study. However, we did perform trans-SNARE complex disassembly assays with αSNAP bearing mutations that disrupt binding of its N-terminal loop to membranes (FS mutation), thus decreasing the effective activity of NSF-αSNAP (Winter et al., 2009) or that disrupt SNARE complex binding (KE mutation) (Zhao et al., 2015). Both mutations impaired the disassembly reaction (new Figures 2D, E), showing that binding of αSNAP to the membranes and to the SNAREs is important for disassembly.

8) The authors claim, in the title as well as the text, that Munc18, Munc13, synaptotagmin, and complexin act synergistically to protect trans-SNARE complexes from NSF/SNAP. However, this is not obvious based on the data presented in Figure 3C/D, where greater-than-additive effects are not observed at least for most combinations. Complexin and synaptotagmin on their own, at the concentration used (but see the next point #9), have only modest activity, which might simply reflect the ability of any binding partner to get in the way of NSF/SNAP. In the absence of calcium, there is a bit of additional protection afforded by complexin (but not by synaptotagmin) when Munc18 and Munc13 are already present.

We fully agree with this criticism. We had misunderstood the meaning of the term synergy and we do not use it in the revised manuscript.

9) The claim that complexin does not prevent disassembly of cis SNARE complexes (Figure 4) is at odds with other results in the literature. Previously, at a SNAP:complexin ratio of ~ 18:1, complexin had no effect (Pabst et al., 2000), whereas at a ratio of ~ 3:1, complexin had a substantial effect on NSF-mediated disassembly of the soluble SNARE complex (Winter et al., 2009). More recently, concentration-dependent complexin inhibition of NSF-mediated disassembly of the cis SNARE complex was observed (Choi et al., 2018). Please perform the experiments shown in Figures 3, 4, 6 at different complexin concentrations, i.e., different complexin:SNAP molar ratios.

We measured the dependence of cis-SNARE complex disassembly by NSF-αSNAP on complexin-1 concentration and αSNAP/complexin-1 ratio (Figure 6—figure supplement 1A, C). We did not see protection under any of these conditions. In the revised manuscript we suggest that differences with respect to the results of these previous papers may arise because our experiments used full-length syntaxin-1 whereas the previous results were obtained with syntaxin-1 lacking the N-terminal region containing the H_abc_ domain; some of the previous results may also have been influenced by the lack of membranes, which renders NSF-αSNAP much less active (subsection “Disassembly of cis-SNARE complexes”, last paragraph).

We also measured the dependence of trans-SNARE complex disassembly on complexin1 concentration, which is now presented in Figure 5—figure supplement 5.

10) Figures 3 and 6 show protection of disassembly of trans SNARE complexes by a variety of factors. A key difference between experiments in these two figures relates to synaptotagmin. The C2AB fragment is used in Figure 6, whereas full-length synaptotagmin is used in Figure 6. Why? Please redo all experiments in Figure 3 with membrane anchored synaptotagmin (as in Figure 6).

We performed the experiments of Figure 3 of the old manuscript with the synaptotagmin-1 C_2_AB fragment so that we could measure the effects of including the fragment or not in the presence or absence of the other proteins. We have repeated these experiments without including Syb29-93 with NSF-αSNAP (now Figure 5B) and in addition performed protection experiments for trans-SNARE complexes formed between liposomes containing synaptobrevin and synaptotagmin-1 (VSyt1-liposomes) and T-liposomes (Figure 5C).

[Editors' note: further revisions were requested prior to acceptance, as described below.]

Major comments:1) The data make a convincing case that complexin can bind trans-SNARE complexes to block NSF/SNAP-driven disassembly. This conclusion, moreover, seems entirely plausible based on previous findings, including high-resolution structures. The role(s) of Munc18/13 on inhibiting trans SNARE complex disassembly is more problematic. It was known that, working together, they powerfully promote trans-SNARE complex assembly. The simplest model, call it Model I, is that Munc18/13 assembly activity simply opposes NSF/SNAP disassembly activity. Model I does not require that Munc18/13 interact with trans-SNARE complexes. By contrast the author's model, call it Model II, is that Munc18/13 remains associated with assembled trans-SNARE complexes. In so doing, Munc18/13 blocks NSF/SNAP from binding, and therefore disassembling, trans-SNARE complexes.Moreover, NSF/SNAP plays a dual role: not only does it catalyze SNARE disassembly, but it also promotes Munc18/13-mediated SNARE assembly by rescuing SNAREs that are kinetically trapped in off-pathway, dead end products. No one, of course, knows this better than the authors, who indeed point out this effect in Figure 4B. Nevertheless, the dual role of NSF/SNAP further complicates the task of unambiguously distinguishing models I and II by using the author's assays.The revised manuscript does not conclusively rule out Model I since every manipulation that strengthens the postulated 'protective' activity of Munc18/13, including the addition of Ca^2+^, also increases its assembly activity. Even the authors, upon occasion, seem to adopt the language of Model I. For example, "[T]he contrast of the results obtained with cis-SNARE complexes with those observed with trans-SNARE complexes provides a dramatic demonstration of how the apposition of two membranes tilts the balance in favor of SNARE complex assembly, whereas disassembly dominates on a single membrane".What would seem to be needed for distinguishing Model I from Model II is some manipulation or mutation that separates the proposed Munc18/13's protective function from its assembly function. For example, this could be accomplished by determining if Munc18 and Munc13 are both associated with trans SNARE complexes prior to NSF/aSNAP-mediated disassembly. However, we realize that such a direct observation may not be possible with the author's assays. At the minimum, we request a more balanced discussion of these points, presenting both models I and II as being consistent with the data presented in this paper and other available data, as well as adjusting the title, Abstract and other relevant parts of the text.

We understand the overall concern about distinguishing between models I and II; this was one of the reasons why in the experiments described in our original manuscript we had used the synaptobrevin SNARE motif (Syb29-93) during our disassembly assays, as the large excess of this soluble fragment would be expected to favor its incorporation into newly formed SNARE complexes, preventing the re-assembly of trans-SNARE complexes. Following the instructions from the first round of review, we removed these data in the revised manuscript and repeated all the experiments without Syb29-93. We believe it is not worth putting all the data that we acquired with Syb29-93 back into the paper, but in the new revised manuscript we did include one experiment where transSNARE complexes were assembled in the presence of Munc18-1, Mun13-1 C_1_C_2_BMUNC_2_C and Ca^2+^, and NSF-αSNAP were added at the end together with an excess of Syb29-93 (new Figure 4—figure supplement 1). The finding that Syb29-93 does not affect the results (compare with Figure 4B) provides strong evidence that disassembly-reassembly of SNARE complex is not occurring substantially under the conditions of these experiments, since we would have observed a gradual decrease in FRET as Syb29-93 becomes incorporated into the complexes.

There are multiple other arguments that support model II against model I, as described in the paragraph that we have included in the Discussion (fourth paragraph) and is copied below. One of these arguments is that αSNAP was reported to strongly inhibit lipid mixing between liposomes by binding to trans-SNARE complexes (Park et al., 2014). Hence, fusion might be arrested if Munc18-1 and Munc13-1 do not prevent αSNAP binding. We have an extensive study of αSNAP function that supports these conclusions and we hope to submit for publication soon. We cannot cite these unpublished results in the paper because this goes against journal policy, but we do cite the data from Park et al. as one of the arguments favoring model II.

We do realize that none of the arguments that we provide is ‘full-proof’ and hence there is a small but reasonable possibility that Model I might be correct. It is too cumbersome to change the wording of the entire Results and Discussion sections to always accommodate the two models, and it might be confusing to the readers. To avoid being too conclusive and to make sure that readers understand the existence of the two possibilities, we have changed the title, the Abstract and some other key parts of the manuscript to make our wording consistent with both models. Moreover, in the new revised manuscript we have included the following warning:

‘Note however that we cannot completely rule out the possibility that, instead of physically preventing disassembly, Munc18-1 and Munc13-1 C_1_C_2_BMUNC_2_C mediate fast re-assembly of trans-SNARE complexes after they are disassembled. For simplicity, below we use terms like ‘prevent’ or ‘protect against disassembly’ to reflect the observation that a particular factor(s) increases the amount of assembled trans-SNARE complexes observed in the presence of NSF-αSNAP, but it is important to keep in mind both possible interpretations (see discussion).’

We still propose model II in last sentence of the Abstract for the reasons explained above and in the paragraph below, but we do not claim that this model has been demonstrated.

We also note that we have re-written some parts of the Discussion to accommodate the paragraph that discusses the merits of model II over model, which reads as follows:

‘An alternative interpretation of these results is that Munc18-1 and Munc13-1 C_1_C_2_BMUNC_2_C do not prevent disassembly but instead mediate fast re-assembly of trans-SNARE complexes after they are disassembled by NSF-αSNAP. […] This model can explain why Ca^2+^ increases the ability of Munc13-1 C_1_C_2_BMUNC_2_C (together with Munc18-1) to protect trans-SNARE complexes against disassembly by NSF-αSNAP (Figure 5B, C).’

2) An additional model to explain the data is as follows (call it Model III): Munc18/13 catalyze the proper assembly of the SNARE complex, and thereby, supercomplexes with complexin and synaptotagmin. Once such properly assembled supercomplexes are situated between membranes, it might be difficult for SNAP/NSF to bind to the trans-SNARE complex even if Munc13/Munc18 are not associated with such supercomplexes anymore. The absence of Munc18/Munc13 may lead to a mixture of properly and improperly assembled trans-SNARE complexes where the latter are more readily disassembled by NSF/SNAP. Please discuss this possible model as well.

We are not sure what the reviewers mean by properly and improperly assembled transSNARE complexes. There is evidence for formation of antiparallel trans-SNARE complexes, but those would not be detected by our FRET assays because our FRET probes right before the N-termini of the synaptobrevin and syntaxin-1 SNARE motifs. Since the Discussion is already very long and we are not sure of the relevance of these ideas to the data that we present, we prefer not discussing them in the paper. We hope that this is acceptable.

3) The authors performed additional experiments to test inhibition of cis-SNARE complexes by complexin (Figure 6, Figure 6—figure supplement 1, Results section "Disassembly of cis-SNARE complexes”; Discussion section, seventh paragraph), and state: "It is plausible that the differences observed with the results of Winter et al.. 2009, and Choi et al., 2018, arose because syntaxin-1 did not include the N-terminal region containing the Habc domain in both of these studies, and in the latter NSF- αSNAP might have been less active because of the absence of membranes." However, contrary to what is stated here, the Winter et al. experiments did include a membrane. The reviewers are not convinced that there is any discrepancy between the data presented in the present work and the previous work by Winter et al., and Choi et al. since the experiment shown in Figure 6—figure supplement 1 does not have the necessary time resolution to discern the effect of complexin on the kinetics of NSF/aSNAP-mediated SNARE complex disassembly observed by Winter et al. and Choi et al.Moreover, the drawing in Figure 6—figure supplement 1 is somewhat misleading since upon injection of NSF/aSNAP, at least a minute (or more) passes before the fluorescence intensity measurements continue. Thus, this break period should be clearly marked in the figure and explained in the figure caption or the Materials and methods. The presentation and discussion of the results in the subsection “Disassembly of cis-SNARE complexes” and the Discussion, should be adjusted to present a unified view of the action of complexin on slowing the kinetics of NSF/aSNAP-mediated for both cis and trans SNARE complexes (i.e., complexin slows the kinetics in both cases to different degrees).

We apologize for not describing some of the details of these experiments and not properly illustrating in the figures the delay occurring between addition of NSF-αSNAP and the moment we re-started monitoring the donor fluorescence. In the new revised manuscript we have modified Figures 4 and 6, as well as Figure 6—figure supplement 1, to illustrate this delay. We have also indicated the existence of this delay in the figure legends, and we give more details in the Materials and methods section (subsection “FRET assays to monitor trans-SNARE complex assembly and disassembly”, second paragraph). We also made it clear in the text that we did not make any attempt to monitor the kinetics of disassembly (subsection “Multiple factors stabilize trans-SNARE complexes against disassembly by NSF-αSNAP”, first paragraph), and that our data are not inconsistent with those of Choi et al., 2018 and Winter et al., 2009 (subsection “Disassembly of cis-SNARE complexes”, last paragraph and Discussion, eighth paragraph).